# Allocating Variance to Maximize Expectation

**Renato Paes Leme** [* 1]  **Cliff Stein** [* 1 2]  **Yifeng Teng** [* 1]  **Pratik Worah** [* 1]

## Abstract

We design efficient approximation algorithms for maximizing the expectation of the supremum of families of Gaussian random variables. In particular, let OPT $:= \max_{\sigma_1, \cdots, \sigma_n} \mathbb{E} \sum_{j=1}^{m} \max_{i \in S_j} X_i$, where $X_i$ are Gaussian, $S_j \subset [n]$ and $\sum_i \sigma_i^2 = 1$, then our theoretical results include:

- We characterize the optimal variance allocation – it concentrates on a small subset of variables as $|S_j|$ increases,
- A polynomial time approximation scheme (PTAS) for computing OPT when $m = 1$, and
- An $O(\log n)$ approximation algorithm for computing OPT for general $m > 1$.

## 1. Introduction

The total accuracy of a machine learning (ML) model is constrained by several factors such as the available training data, model architecture, and training algorithms, and while it may not be possible to improve the average model accuracy past a certain point in a given setup, one still has freedom to tune the model to trade-off accuracy in one subset of examples for accuracy in another subset. One approach is to reweigh the data (Quionero-Candela et al., 2009; Cortes et al., 2008; Awasthi et al., 2024).

One interesting example of this situation arises in auctions markets. The goal of these markets is to maximize utility of the goods to be allocated. Assuming that bidders in auctions bid true to their utility, the first price auction chooses the maximum valued bidder to allocate the goods. When we have a probabilistic model for the bidder's utility then we can optimize the expected utility by maximizing the expectation of the auction settling price – the maximum utility of the bidders. In the setting when the bidder utilities can be modeled using simple linear regression or via wide neural nets, the expected utility maximization problem reduces to an expectation maximization problem over Gaussian process, as the model training dynamics are described by Gaussian processes. Moreover, the error from each bidder model is directly related to the variance of the underlying Gaussian. Since training the bidder models takes time and requires a large number of samples, we have a natural budget on the total variance over all bidder models. Therefore, in the above setting, the utility maximization problem is equivalent to the problem of designing efficient algorithms to optimally allocate variance for a simple class of objective functions via an expectation maximization problem, where the expectation is over the maximum of some Gaussians.[1] In this paper, we concretely formulate and solve different versions of the aforementioned utility maximization problem, corresponding to different levels of bidder interaction, by designing polynomial time approximation schemes (PTAS).

The organization of the paper is as follows. In Subsection 1.1, we formulate the abstract optimization problems formally; in Subsection 1.2, we state our main structural lemma and theorems, which correspond to the proofs of correctness of the approximation algorithm mentioned above; in Subsection 1.3, we provide figures (Figures 1 and 2) that show general trends in the solution as the dependency among the Gaussians increases; in Subsection 1.4 we discuss applications to machine learning and auctions; in Subsection 1.5, we discuss related works, especially how our results compare to prior extensive work in this area (Talagrand, 2021); and finally in Section 2 we describe an overview of our theorems and prove Lemma 2.1, which contains a modified chaining argument that may be of independent interest. The appendix contains the remaining theorems and proofs.

### 1.1. Problem formulation

We are motivated by trade-offs in the accuracy of ML models, but we defer any motivational details to Subsection 1.4 and formulate our main question as a stochastic optimization problem. We start by defining its basic version (dense

---

[1]Google Research, New York, NY, United States [2]IEOR and Computer Science Department, Columbia University, New York, NY, United States. Correspondence to: Yifeng Teng <yifengt@google.com>.

*Proceedings of the 43rd International Conference on Machine Learning*, Seoul, South Korea. PMLR 306, 2026. Copyright 2026 by the author(s).

---

[1]We discuss a more concrete version of this motivation in Subsection 1.4.

and independent) and later consider correlated and more general graph variants.

*Variance Allocation Problem:* We are given $n$ independent Gaussian random variables $(X_1, \ldots, X_n)$ with means $\mu_1, \ldots, \mu_n$. Our goal is to assign values to the variances, $\sigma_1^2, \ldots, \sigma_n^2$, so as to maximize the expected value of the maximum $X_i$ subject to the constraint that the total variance is bounded, that is the variance vector satisfies $\sum_{i=1}^{n} \sigma_i^2 = 1$. We let OPT denote the optimal objective in this setting, and summarize the problem as (VarAlloc):

$$\text{OPT} \quad = \quad \max_{\sigma_1, \ldots, \sigma_n} \mathbb{E} \max_{i \in [n]} X_i \qquad (1)$$

$$\text{subject to} \qquad \sum_{i=1}^{n} \sigma_i^2 = 1. \qquad (2)$$

*Correlated Variance Allocation:* The above formulation (VarAlloc) uses independent Gaussians, but we also consider the non-independent case, where we have covariances. We are again given means $\mu_1, \ldots, \mu_n$ and are asked to choose a covariance matrix $\Sigma$ to maximize the expected maximum of $(X_1, X_2, \cdots, X_n) \sim \mathcal{N}(\mu, \Sigma)$ such that the total variance is bounded $\sum_{i=1}^{n} \Sigma_{ii} = 1$. Thus we get (CorrVarAlloc):

$$\text{OPT} \quad = \quad \max_{\Sigma} \mathbb{E} \max_{i \in [n]} X_i \qquad (3)$$

$$\text{subject to} \qquad \sum_{i=1}^{n} \Sigma_{ii} = 1 \qquad (4)$$

$$\Sigma \text{ is PSD}. \qquad (5)$$

*Graph Variance Allocation:* We generalize (VarAlloc) in a different way by allowing grouping among variables. We are now given, in addition to the means $\mu_1, \ldots, \mu_n$ of $n$ independent and normally distributed variables, $m$ subsets $S_j \subseteq [n]$. We denote the contribution of the set $S_j$ by: $\max_{i \in S_j} X_i$, and our objective is to allocate the variance with the same constraint (2) and maximize the total expected contribution of the $m$ sets. More formally (GraphVarAlloc):

$$\text{OPT} \quad = \quad \max_{\sigma_1, \cdots, \sigma_n} \mathbb{E} \sum_{j=1}^{m} \max_{i \in S_j} X_i \qquad (6)$$

$$\text{subject to} \qquad \sum_{i=1}^{n} \sigma_i^2 = 1. \qquad (7)$$

*Correlated Graph Variance Allocation:* We can also generalize (GraphVarAlloc) to use correlated Gaussians. We are again given means $\mu_1, \ldots, \mu_n$ and $m$ subsets $S_j \subseteq [n]$, and want to choose a covariance matrix $\Sigma$ to maximize the sum of the contribution $\max_{i \in S_j} X_i$ of each set $S_j$ such that the total variance is bounded $\sum_{i=1}^{n} \Sigma_{ii} = 1$. More formally (CorrGraphVarAlloc):

$$\text{OPT} \quad = \quad \max_{\Sigma} \mathbb{E} \sum_{j=1}^{m} \max_{i \in S_j} X_i \qquad (8)$$

$$\text{subject to} \qquad \sum_{i=1}^{n} \Sigma_{ii} = 1 \qquad (9)$$

$$\Sigma \text{ is PSD}. \qquad (10)$$

Note that, when $\Sigma$ is diagonal:

- CorrVarAlloc is equivalent to VarAlloc, and

- CorrGraphVarAlloc is equivalent to GraphVarAlloc;

and when $m = 1$, GraphVarAlloc is equivalent to VarAlloc. In each case, we want to compute the optimal $\boldsymbol{\sigma}$s or $\Sigma$ using an efficient approximation algorithm. We will use OBJ to denote the objective function studied in each case: $\text{OBJ} = \mathbb{E} \max_{i \in [n]} X_i$ for VarAlloc and CorrVarAlloc, and $\text{OBJ} = \mathbb{E} \sum_{j=1}^{m} \max_{i \in S_j} X_i$ for GraphVarAlloc.

### 1.2. Main algorithmic results

We state our main technical results here. Our algorithms for VarAlloc and CorrVarAlloc will be stated as additive approximations, while for GraphVarAlloc, it is stated a multiplicative approximation. For VarAlloc and CorrVarAlloc, the additive PTAS is at least as strong as a multiplicative PTAS, since OPT is at least $\Omega(1)$ even when $\mu_i = 0$ for every $i$: when setting $\sigma_1 = 1$ and $\sigma_i = 0$ for $i \geq 2$, $\mathbb{E} \max_i X_i = \frac{1}{\sqrt{2\pi}}$ for $n \geq 2$ and $0$ for $n = 1$.

**Theorem 1.1.** *Given an input to* VarAlloc *with* $\mu_1, \cdots, \mu_n \geq 0$ *and constant* $\epsilon > 0$*, there exists an algorithm with running time polynomial in* $n$*, that computes a variance vector* $(\hat{\sigma}_1, \cdots, \hat{\sigma}_n)$ *such that* $\mathbb{E} \max_{i \in [n]} X_i \geq \text{OPT} - \epsilon$.

**Theorem 1.2.** *Given an input to* CorrVarAlloc *and constant* $\epsilon > 0$*, there exists an algorithm with running time polynomial in* $n$*, that computes a covariance matrix* $\widehat{\Sigma}$ *such that for* $(X_1, \cdots, X_n) \sim \mathcal{N}(\mu, \widehat{\Sigma})$*,* $\mathbb{E} \max_{i \in [n]} X_i \geq \text{OPT} - \epsilon$.

**Theorem 1.3.** *Given an input to* GraphVarAlloc*, there exists an algorithm with a running time polynomial in* $n$ *that computes a variance vector* $(\hat{\sigma}_1, \cdots, \hat{\sigma}_n)$ *such that* $\mathbb{E} \sum_{j=1}^{m} \max_{i \in S_j} X_i \geq \Omega\left(\frac{1}{\log n}\right) \text{OPT}$.

**Theorem 1.4.** *Given an input to* CorrGraphVarAlloc*, there exists an algorithm with running time polynomial in* $n$ *that computes a covariance matrix* $\widehat{\Sigma}$ *such that for* $(X_1, \cdots, X_n) \sim \mathcal{N}(\mu, \widehat{\Sigma})$*,* $\mathbb{E} \sum_{j=1}^{m} \max_{i \in S_j} X_i \geq \Omega\left(\frac{1}{\log n}\right) \text{OPT}$.

The main proof ideas and algorithms are discussed in Section 2. For complete proofs, the optimization problem with a single set is discussed in Section A. The proof of Theorem 1.1 is discussed in Section A.1. The proof of Theorem 1.2 is discussed in Section A.2. The optimization problem with multiple sets is discussed in Section B. The proofs of Theorem 1.3 and Theorem 1.4 are discussed in Section B.2. As we do not prove the tightness of our algorithms, we leave the improvement of our algorithms as open questions.

### 1.3. Properties of optimal solution

In order to understand the structure of the optimal variance allocations, we run Monte-Carlo simulations on Erdós-Renyi random graphs and obtain the following plots (Figures 1 and 2) that characterize the optimal value (OPT) and the corresponding variance allocation for GraphVarAlloc. We found OPT to be concave, and the corresponding allocation to be concentrated on a few variables, as the density of the instance increased – a somewhat counterintuitive pattern – even for small random graphs (see Figures 1 and 2):

- *Concavity:* The solution value OPT appears to be concave as a function of $p = \frac{|S_j|}{n}$, irrespective of whether the Gaussian random variables are restricted to be independent, positively or negatively correlated (see Figure 1). This concavity is supported by Theorem 1.5, which shows that the objective for GraphVarAlloc, as a function of $p$ (where $p = \frac{|S_j|}{n}$), is strictly concave for any fixed variance allocation. The key to the proof is the submodularity of the objective function.

- *Concentration:* The variances tend to concentrate, i.e., they are supported on a smaller subset of variables with increasing $p$, where $p = \frac{|S_j|}{n}$ (see Figure 2(a,b,c,d)). This concentration is formally verified in Theorem 1.6, which shows that only $\Theta(\frac{1}{p})$ variables are allocated a variance $\Omega(p)$ in the optimal allocation, i.e., most variances are small in the optimal allocation as $p$ increases. The proof of this theorem relies on Lemma 2.1, which is our key structural lemma, and also is used in the proof of correctness of our approximation algorithms (c.f. Subsection 1.2).

**Theorem 1.5.** *Let $I_k$ be an instance for GraphVarAlloc with all $\binom{n}{k}$ sets of size $k$: for every $1 \le i_1 < i_2 < \cdots < i_k \le n$, there is a set $S_j = \{i_1, \cdots, i_k\}$ in this instance. For any fixed variance allocation $\boldsymbol{\sigma}$, let $f_{\boldsymbol{\sigma}}(k) = \frac{1}{\binom{n}{k}} \mathrm{OBJ}_{I_k}(\boldsymbol{\sigma})$ be the expected per-set contribution. Then $f_{\boldsymbol{\sigma}}(k)$ is a concave function with respect to $k$.*

**Theorem 1.6.** *For any $\delta > 0$ and a random instance of GraphVarAlloc with $n, m \to \infty$, with probability at least $1 - \delta$, there are $\Theta(\frac{1}{p})$ variables allocated variance $\Omega(p)$.*

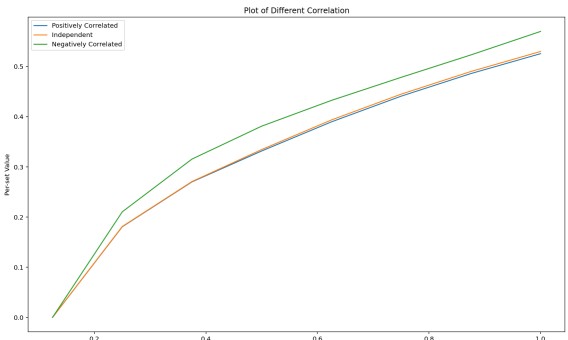

*Figure 1.* Concavity of optimal objective value per set (i.e. $\frac{1}{m}\mathrm{OPT}$) when the Gaussians are restricted to be independent, be positively, or be negatively correlated in GraphVarAlloc, as a function of $p$ for Erdós-Renyi graphs with $p \in \{\frac{1}{8}, \frac{2}{8}, ..., \frac{8}{8}\}$ and $n = 8$. To improve the simulation speed the positive and negative correlations are limited to block-diagonal matrices with block sizes $2 \times 2$, and $\Sigma_{2i+1, 2i+2} = \pm\sqrt{\Sigma_{2i+1, 2i+1} \cdot \Sigma_{2i+2, 2i+2}}$. See also Theorem 1.5 for the discussion of the independent case.

### 1.4. Motivating application

It is well-known that the study of bounds on the maximum value of stochastic processes $\mathbb{E}[\max_t X_t]$ has important applications, both practical ones ("What is the maximum magnitude of an earthquake in this region?" or "What is the maximum level a river will rise?"), and theoretical ones, such as a relation between properties of metric spaces and bounds on stochastic processes. The theoretical applications are discussed in detail in (Talagrand, 2021). In this subsection, we outline a motivating application for our problem, where we have an additional budget constraint on the variance.

**Stochastic Gradient Descent**  The study of upper bounds on the expected behavior of stochastic processes plays a vital role in the training of ML models. The basic stochastic gradient descent procedure (SGD), used for training ML models, can be linearized near the local minima,[2] and viewed in the limit of small step size as an Ornstein Uhlenbeck (OU) process, which is a Ito diffusion of the form:

$$dX_t = W(\mu - X_t)dt + A dB_t, \qquad (11)$$

where $X_t, \mu \in \mathbb{R}^n, W, A \in \mathbb{R}^{n \times n}$ and $B_t$ is $n$-dimensional Standard Brownian Motion (see (Oksendal, 2013) for details). When $W$ is a diagonal matrix, it corresponds to changing the relative weight of different features. The long-term distribution of $X$ (obtained after many steps of SGD), can be shown to be a Gaussian distribution.

---

[2]This is globally true for linear regression trained using SGD, but this also locally true near local minima for wide neural networks. In the latter case, the training dynamics are known to be a Gaussian process in the limit of large width (Gao et al., 2023).

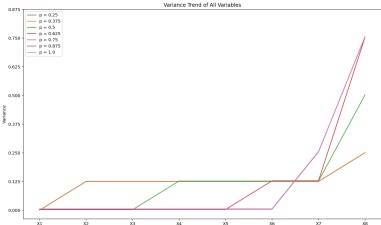

*(a)* Optimal variance allocation for non-zero mean independent Gaussians, where the mean vector is $(0, 0, 0, 0, 0, 0, 1, 1)$

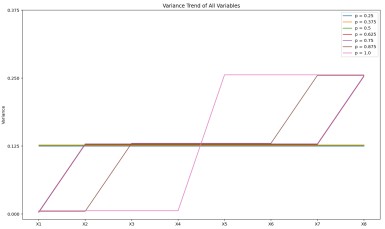

*(b)* Optimal variance allocation for zero mean independent Gaussians

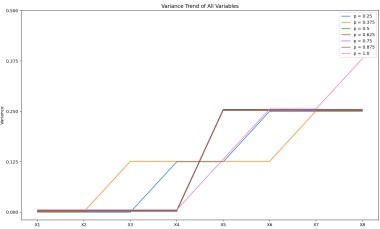

*(c)* Optimal variance allocation for zero mean positively correlated Gaussians

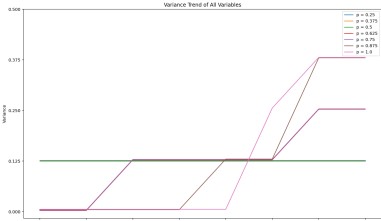

*(d)* Optimal variance allocation for zero mean negatively correlated Gaussians

*Figure 2.* Simulations for Erdós-Renyi graphs $G(n, p)$ for GraphVarAlloc ($n$ is held constant). $x$-axis corresponds to the $n$ variances (sorted by increasing values) and $y$-axis corresponds to their optimal values in allocation. Note the increasingly concentrated variance allocation with increasing $p$. See also Theorem 1.6.

**Case Study: Auction and Learning**  Let's consider a more specific problem setting to better motivate our problem in the learning setting. Consider an online advertising environment and let each variable $x_i$ be the prediction of the click-through-rate (CTR) of a given bidder $i$. Usually, the ad platform trains a single model for each bidder and, given values $v_i$ for a click, the objective function $\mathbb{E}[\max_i v_i x_i]$ corresponds to the expected revenue in a first price auction ($n$ bidders participating in one auction). Suppose one wants

to compute optimal bids $y \in \mathbb{R}^n$, using conversion probabilities $x \in (0, 1)^n$, for $n$ independent bidders in a first price auction for click ads, via least-squares regression. So $y \approx \beta x$. Further assume that:

- The terms in the least-square loss function are weighted by a PSD matrix $W$[3], which we can control. To ensure that the model converges to the optimum for all bidders, it suffices to require that $W$ is positive definite.[4]

- We use SGD with a small step-size to find the matrix $\beta \in \mathbb{R}^{n \times n}$.

Furthermore, just for simplicity, let's assume $\beta$ and $W$ are diagonal. Then, assuming that mean of $y = \mu$ and $Var(x_i) = \sigma_i^2$ the training dynamics are:

$$d\beta_{ii}(t) = W_{ii}(\mu_i - \sigma_i^2 \beta_{ii}(t))dt + AdB_t,$$

where $B_t$ is the standard Brownian motion and $A$ is the batch variance (assumed constant for simplicity). Observe that, as $t \to \infty$, $\beta_{ii} \to \frac{\mu_i}{\sigma_i^2}$, the speed of convergence is $W_{ii}$, and the distribution of $\beta_{ii}$ is a Gaussian:

$$\mathcal{N}\left(\frac{\mu_i}{\sigma_i^2}, \frac{A^2}{2W_{ii}}\right). \tag{12}$$

Since $W$ is diagonal, positive definiteness implies: $\forall i : W_{ii} \geq \varepsilon_i > 0$. This implies $\frac{1}{W_{ii}} \leq \frac{1}{\varepsilon_i}$. The latter together with Equation 12, implies an upper bound on the total variance:

$$\sum_i \frac{A^2}{2W_{ii}} = A^2 \cdot \sum_i \frac{1}{W_{ii}} \leq \frac{A^2 \cdot n}{\min_i \varepsilon_i}. \tag{13}$$

Since diffusivity $A^2$ is assumed small and positive (small step size), the upper bound can be assumed to be constant when it is small enough: $A = \Theta(\frac{1}{\sqrt{n}})$. This implies a budget for the total variance. Since we are interested in the relative allocation of the variances, we scale the total budget to 1.

Now, if we want to maximize the revenue of our auction, then we need to maximize $\mathbb{E}[\max_i \{\beta_{ii}\}]$ over $W$s above.[5] This is precisely the basic version of our problem – VarAlloc!

In summary, to maximize revenue in a first price auction, we want to upweight the contribution of bidders that have

---

[3]$W$ is the speed matrix of the OU process, which determines the speed of mean reversion.

[4]Note that without positive definiteness, negative eigenvalues would lead the OU process to diverge in the direction of the corresponding eigenvector.

[5]The max inside the expectation arises from the objective function in first price auctions.

a higher probability of conversions (clicks) without affecting the overall loss too much, and our variance allocation problem formulates one way of doing that.

When the bidders may depend on each other, for example, they may be anticipating market demand and competing, then the covariance matrices are no longer diagonal. The long-term distribution of $\beta$ will remain a Gaussian, and let $\Sigma$ be the covariance matrix of $\beta$; then: $\Sigma W + W^T \Sigma = AA^T$. Now, to maximize revenue, one needs to maximize $\mathbb{E}[\max_i\{\beta_{ii}x_i\}]$ over $W$s, under the constraint in the previous equation.[6] Note $\Sigma$ and $A$ are given. CorVarAlloc is our formulation for this problem.

The above discussion is for $n$ bidders participating in one auction. GraphVarAlloc models the case when there are multiple auctions, but each bidder may only participate in some of the auctions. The underlying graph models the market structure, and we use models on $G(n, p)$ to study what happens when the graph representing auction participation is random. Here, the objective measures the average revenue over all first price auctions in the market. We found that revenue behaves in a concave manner (Figure 1 in our paper), and the graphs in Figure 2 study what kind of bidding patterns maximize revenue in various market structures. We also provide approximation results for general underlying market structures that correspond to the model CorrGraphVarAllocwith correlated variables and arbitrary graphs.

### 1.5. Related work

Talagrand (Talagrand, 2021) has extensively studied problems of the form $\mathbb{E}[\sup_{t \in T}\{X_t\}]$, where $X_t$ are Gaussian. However, we are not aware of any of his work for the corresponding problems VarAlloc, CorrVarAlloc, GraphVarAlloc, or CorrGraphVarAlloc, where we have (1) a budget on the total variance of the underlying Gaussian variables, and (2) the variables may have non-zero means. That said, our main structural lemma (Lemma 2.1) does rely on a chaining argument (as do many other results in this area), which has been discussed thoroughly in Chapter 2 of (Talagrand, 2021). However, a direct application of methods in Chapter 2 of (Talagrand, 2021) would give an upper bound of $O(\epsilon\sqrt{\log n})$ as opposed to $O(\epsilon\sqrt{\log \frac{1}{\epsilon}})$ (cf. Lemma 2.1).

Our approximation algorithms corresponding to Theorems 1.2 and 1.3 rely crucially on Lemma 2.1. We are able to obtain a PTAS because Lemma 2.1 implies that most of the variables will be set to zero in the optimal solution (a

version of that result is in Theorem 1.6). The logarithmic approximation algorithm, on the other hand, is from a reduction to submodular maximization, which is NP-hard but admits an efficient approximation algorithm (Nemhauser et al., 1978). As an aside, it's worth noting that many variants of submodular maximization have been studied before (Vondrák, 2013), including for non-monotone submodular functions (Buchbinder & Feldman, 2018), monotone submodular functions (Feige et al., 2011), and submodular maximization with a matroid constraint (Buchbinder & Feldman, 2024).

## 2. Overview of technical results and proofs

In this section we provide an overview of the proofs of Theorems 1.1, 1.2 and 1.3 from Section 1.2. The proof of Lemma 2.1, which may be of independent interest, is presented in this section as well. The remaining detailed proofs are deferred to the appendix.

### 2.1. Expectation maximization for a single set with independent variables

As the optimization problems underlying VarAlloc, CorrVarAlloc, and GraphVarAlloc are non-convex, even when there is a single set, it is hard to find efficient algorithms to solve the problems exactly. Although the solution space of the problem has $n$ dimensions for the independent setting and $n^2$ dimensions for joint Gaussian variables, it is still possible to find a polynomial-time approximation scheme, either additive or multiplicative.

The main intuition for the PTAS is as follows. We want to reduce the search space of near-optimal solutions. Firstly, all random variables with variance at most $\epsilon^2$ can only contribute to $\tilde{O}(\epsilon)$ in the optimization objective.[7] Such an observation is formally shown by Lemma 2.1. Therefore, we only need to consider solutions with positive variances of the variables being large enough ($\geq \epsilon^2$), and there can only be $O(\epsilon^{-2})$ such variables, which is a constant for any fixed $\epsilon$.

**Lemma 2.1.** *For any $\epsilon \in (0,1)$ and positive integer $m$, let $m$ zero-mean Gaussian random variables $(Y_1, Y_2, \cdots, Y_m) \sim \mathcal{N}(0, \Sigma)$ satisfy $0 \leq \Sigma_{11}, \Sigma_{22}, \cdots, \Sigma_{mm} \leq \epsilon^2$ and $\sum_i \Sigma_{ii} \leq 1$, we have $\mathbb{E}\max(0, \max_{i \in [m]} Y_i) = O(\epsilon\sqrt{\ln \frac{1}{\epsilon}})$.*

We observe that a weaker bound of $O(\epsilon\sqrt{\log n})$ follows from directly applying Talagrand's chaining arguments in Chapter 2 of (Talagrand, 2021). The weaker bound leads to a QPTAS (i.e. it runs in $n^{\text{polylog}(n)}$ time) since we can only guarantee polylog$(n)$ variables with negligible vari-

---

[6]Note that for a given optimal solution $\Sigma$ (and diffusivity $A$), we need to find a positive definite $W$ such that a solution exists to the equality. This is the degree of fine-grained control we can have over covariances.

[7]$\tilde{O}(\cdot)$ means the standard big O notation ignoring any polylogarithmic factor.

ance. Here we are able to obtain the stronger bound of $O(\epsilon\sqrt{\log\frac{1}{\epsilon}})$, which is essential to the PTAS.

Our proof begins by following the approach in (Talagrand, 2021) of approximating the maximum as a sum over maximums of smaller subsets of random variables. However, crucially, since the total variance is bounded, we are able to prove tighter upper bounds on the contribution of each group (Equation 15). The upshot of these improved bounds is to replace a $\log n$ factor by a $\log\frac{1}{\epsilon}$.

*Proof of Lemma 2.1.* Firstly, we group these random variables into groups $G_1, G_2, \cdots$ according to their variance. Each group $G_j$ contains all random variables $Y_i$ with variance $\Sigma_{ii}$ satisfying $2^{-j}\epsilon < \sqrt{\Sigma_{ii}} \leq 2^{1-j}\epsilon$. For variables in group $G_j$, let $Z_j = \max(0, \max_{Y_i \in G_j} Y_i)$ be the maximum of all variables in this group (lower-bounded by 0). Let $m_j = |G_j|$ be the number of variables in this group. We have for any $t \geq 0$,

$$
\begin{aligned}
e^{t\mathbb{E}Z_j} &\leq \mathbb{E}[e^{tZ_j}] = \mathbb{E}[e^{t\max(0, \max_{Y_i \in G_j} Y_i)}] \\
&= \mathbb{E}\max\left(e^{t\cdot 0}, \max_{Y_i \in G_j} e^{tY_i}\right),
\end{aligned}
$$

by the convexity of the exponential function and Jensen's inequality and the definition of $Z_j$. Now, since the expectation of maximum is at most the sum we have:

$$
e^{t\mathbb{E}Z_j} \leq 1 + \sum_{Y_i \in G_j}\mathbb{E}[e^{tY_i}] = 1 + \sum_{Y_i \in G_j} e^{t^2\Sigma_{ii}/2} \leq 1 + m_j e^{2^{1-2j}t^2\epsilon^2},
$$

where the equality follows by the expectation of the exponential of normal variables and the last inequality by $\Sigma_{ii} \leq 2^{2-2j}\epsilon^2$ for any $Y_i \in G_j$. Taking the logarithm of both sides, we get

$$
\begin{aligned}
t\mathbb{E}Z_j &\leq \ln(1 + m_j e^{2^{1-2j}t^2\epsilon^2}) \leq \ln((1+m_j)e^{2^{1-2j}t^2\epsilon^2}) \\
&= \ln(1+m_j) + 2^{1-2j}t^2\epsilon^2.
\end{aligned}
$$

By taking $t = \frac{2^{j-1}\sqrt{2\ln(1+m_j)}}{\epsilon}$, we have

$$
\mathbb{E}Z_j \leq 2^{1-j}\epsilon\sqrt{2\ln(1+m_j)}. \tag{14}
$$

As the total variance of all variables is upper-bounded by 1, we know that

$$
1 \geq \sum_{Y_i \in G_j}\Sigma_{ii} > \sum_{Y_i \in G_j} 2^{-2j}\epsilon^2 = 2^{-2j}\epsilon^2 m_j. \tag{15}
$$

Then we have $m_j < 2^{2j}\epsilon^{-2}$, and applying to (14) we get

$$
\begin{aligned}
\mathbb{E}Z_j &\leq 2^{1-j}\epsilon\sqrt{2\ln(1+2^{2j}\epsilon^{-2})} < 2^{1-j}\epsilon\sqrt{2+2\ln(2^{2j}\epsilon^{-2})} \\
&= 2^{1-j}\epsilon \cdot \left(\sqrt{j} + \sqrt{\ln\frac{1}{\epsilon}}\right) \cdot O(1). \tag{16}
\end{aligned}
$$

Here the second inequality comes from $\ln(1+x) < 1 + \ln x$ for $x > 1$; the last equality comes from extracting the dominant terms below the square root. By taking the max of $Z_j$ over all groups, we have

$$
\begin{aligned}
\mathbb{E}\max(0, \max_{i \in [m]} Y_i) &= \mathbb{E}\max_j Z_j \leq \sum_j \mathbb{E}Z_j \\
&< \sum_j 2^{1-j}\epsilon\left(\sqrt{j} + \sqrt{\ln\frac{1}{\epsilon}}\right) \cdot O(1),
\end{aligned}
$$

where the last inequality follows from (16). Finally, notice that the sum with $\sqrt{j}$ terms is $O(\epsilon)$, which has a lower order compared to the sum with $\sqrt{\ln\frac{1}{\epsilon}}$ terms, from which we conclude that:

$$
\mathbb{E}\max(0, \max_{i \in [m]} Y_i) \leq O\left(\epsilon\sqrt{\ln\frac{1}{\epsilon}}\right).
$$

$\square$

Secondly, for any two vectors of independent normal variables with small $\ell_1$ distance in variances, their respective expected maximum values are close. Thus to search for a near-optimal solution, it suffices to do a grid search on all possible variance vectors with each positive element being an integral multiple of $poly(\epsilon)$ (we use $\epsilon^{-3}$ in Algorithm 1). Any variance vector not being a grid point has a similar objective due to Lemma 2.2.

**Lemma 2.2.** *Consider two vectors of standard deviations $(\sigma_1, \cdots, \sigma_n)$ and $(\sigma'_1, \cdots, \sigma'_n)$. Let $X_1, \cdots, X_n$ and $Y_1, \cdots, Y_n$ be two sets of mutually independent normal variables with $X_i \sim \mathcal{N}(\mu_i, \sigma_i^2)$ and $Y_i \sim \mathcal{N}(\mu_i, \sigma'^2_i)$, then*

$$
|\mathbb{E}\max_{i \in [n]} X_i - \mathbb{E}\max_{i \in [n]} Y_i| \leq O(1) \cdot \sum_{i \in [n]} |\sigma_i - \sigma'_i|.
$$

### 2.2. Expectation maximization for a single set with correlated variables

When all random variables come from a multi-dimensional Gaussian distribution, we need to have a different algorithm. While Lemma 2.1 was already shown for correlated Gaussian variables, which means that we only need to search for covariance matrices with $poly(\frac{1}{\epsilon})$ positive dimensions, the smoothness condition from Lemma 2.2 no longer works. Instead, we observe that for any two vectors of random variables with a small Earth Mover's distance, their respective expected maximum values are close.[8] Another observation

---

[8]The Earth Mover's distance of two measures $\mu, \nu$ on a metric space $(X, d)$ is defined to be $\inf_{\pi \in \Pi(\mu,\nu)} \int_{X \times X} d(x, y) d\pi(x, y)$, where $\Pi$ is the set of all couplings of $\mu$ and $\nu$.

**Algorithm 1** A PTAS algorithm distributing variances to independent arbitrary-mean variables

1: **Input:** $\mu_1, \mu_2, \cdots, \mu_n \geq 0$
2: $k \leftarrow \frac{1}{\epsilon^2}$;
3: Set current maximum $M = 0$;
4: **for** all possible $k$ indices $1 \leq i_1 < i_2 < \cdots < i_k \leq n$ **do**
5:     **for** any $i \notin \{i_1, \cdots, i_k\}$ **do**
6:         Set $\hat{\sigma}_i = 0$;
7:     **end for**
8:     **for** all possible $(\sigma_{i_1}, \cdots, \sigma_{i_k})$, where $\sigma_{i_j}$ is an integral multiple of $\epsilon^3$ for every $j \in [k]$ **do**
9:         **if** $\sum_{j=1}^{k} \sigma_{i_j}^2 \leq 1$ **and** $\mathbb{E} \max_{i \in [n]} X_i > M$ **then**
10:           $(\hat{\sigma}_{i_1}, \cdots, \hat{\sigma}_{i_k}) \leftarrow (\sigma_{i_1}, \cdots, \sigma_{i_k})$;
11:           $M \leftarrow \mathrm{OBJ}(\boldsymbol{\sigma})$;
12:         **end if**
13:     **end for**
14: **end for**
15: **Output:** $\hat{\boldsymbol{\sigma}} = (\hat{\sigma}_1, \cdots, \hat{\sigma}_n)$

**Algorithm 2** A PTAS algorithm distributing variances to arbitrary-mean correlated variables

**Input:** $\mu_1, \mu_2, \cdots, \mu_n \geq 0$
$k \leftarrow \frac{1}{\epsilon^2}$;
Set current maximum $M = 0$;
**for** all possible $k$ indices $1 \leq i_1 < i_2 < \cdots < i_k \leq n$ **do**
    **for** any $i \notin \{i_1, \cdots, i_k\}$ **do**
        **for** any $\ell \in [n]$ **do**
          set $\widehat{\Sigma}_{i\ell} = \widehat{\Sigma}_{\ell i} = 0$;
        **end for**
    **end for**
    **for** any $i, j \in \{i_1, \cdots, i_k\}$ **do**
        **for** all possible values of $(\Sigma_{ij})$ being an integral multiple of $\epsilon^8$ with value in [-1,1] **do**
          **if** $\Sigma$ is positive semi-definite, $\mathrm{tr}(\Sigma) \leq 1$, **and** $\mathrm{OBJ}(\Sigma) > M$ **then**
            $\widehat{\Sigma} \leftarrow \Sigma$;
            $M \leftarrow \mathrm{OBJ}(\Sigma)$;
          **end if**
        **end for**
    **end for**
**end for**
**Output:** $\widehat{\Sigma}$

is that for two multi-dimensional Gaussian random vectors, if the square roots of their covariance matrices are close in entry-wise $\ell_1$ distance, then they have a small Earth Mover's distance.

**Lemma 2.3.** *For two random vectors $X = (X_1, X_2, \cdots, X_n)$ and $Y = (Y_1, Y_2, \cdots, Y_n)$ drawn from distribution $D_X$ and $D_Y$ respectively, if the Earth Mover's Distance $d_{EMD}(D_X, D_Y)$ (defined by $\ell_1$ norm) between two distributions is at most $\epsilon$, then $|\mathbb{E} \max_i X_i - \mathbb{E} \max_i Y_i| \leq \epsilon$.*

**Lemma 2.4.** *For any two dimension-$k$ Gaussian distributions with $D_1 = \mathcal{N}(\mu, A)$ and $D_2 = \mathcal{N}(\mu, B)$, suppose that covariance matrices $A$ and $B$ are close in entry-wise $\ell_1$ distance: $\|A - B\|_{1,1} = \sum_{i,j} |A_{ij} - B_{ij}| \leq \epsilon$, with $\mathrm{tr}(A) = \mathrm{tr}(B) = 1$. Then the Earth Mover Distance between $D_1$ and $D_2$ is upper bounded by $\sqrt{k}\epsilon$.*

Combining the previous observations, to search for a near-optimal solution, it suffices to do a grid search on the covariance matrices with each element being an integral multiple of poly$(\epsilon)$. We can design an efficient algorithm (Algorithm 2) that obtains a near-optimal solution with $O(\epsilon)$ loss for any fixed $\epsilon > 0$. The algorithm performs grid search on $O(\epsilon^{-2})$ of the variables with each variable having variance being multiples of poly$(\epsilon)$ (we use $\epsilon^8$ here compared to the $\epsilon^3$ in the independent case due to a slightly worse smoothness lemma).

### 2.3. Expectation maximization for multiple sets

When there are multiple sets, the problem is much more complicated. The PTAS technique for the single-set setting no longer works, as the scale of the optimal objective can

be very different. For example, when each set has small cardinality, the optimal way to distribute the variance may be to uniformly split the variance budget to all variables, which means that all variables have $O(\frac{1}{n^2})$ variance. The previous grid search algorithm will no longer be efficient as there can be $\Omega(n)$ variables with small variance and contribute significantly to the optimal objective. One example is when each set contains two variables and the sets form a cycle graph.

**Theorem 2.5.** *Suppose $m = n$, and $S_j = \{X_j, X_{j+1}\}$ for every $j \in [n]$ (assume that $X_{n+1} = X_1$). If all variables have the same mean, an optimal way to distribute variance is to set $\sigma_i^2 = \frac{1}{n}$ for each $i$.*

For the most general case of the problem, we obtain an efficient approximation algorithm with a competitive ratio $O(\log n)$. The main intuition is as follows. Firstly, for any solution, variables with tiny variance $< \frac{1}{n}$ only have a negligible contribution to the total objective. For the rest of the variables, we can group the variables according to their variances such that each group contains variables with variance differences bounded by a constant factor. Then since there are only $O(\log n)$ groups, one group of variables contributes $\Omega(\frac{1}{\log n})$ fraction of the optimal objective. Secondly, when the standard deviation of each variable is perturbed by a factor of 2, the overall objective also only changes by at most a factor of 2. This means that if we can solve the problem with each variable having a variance of either 0

or some fixed value, we can obtain $\Omega(\frac{1}{\log n})$ fraction of the optimal objective.

**Lemma 2.6.** *For $n$ zero-mean normally distributed variables $X_1, X_2, \cdots, X_n$ with standard deviation $\sigma_1, \cdots, \sigma_n$, suppose that there are $n$ other zero-mean normally distributed variables $X_1', X_2', \cdots, X_n'$ with standard deviation $\sigma_1', \cdots, \sigma_n'$ satisfying $\sigma_i \leq \sigma_i' \leq 2\sigma_i$, then $\mathbb{E} \max X_i \leq \mathbb{E} \max X_i' \leq 2\mathbb{E} \max X_i$.*

The last observation is that, the problem with each variable having a variance being either 0 or some fixed value is actually a submodular maximization problem (Lemma 2.7), and can be solved efficiently with an approximation factor $1 - \frac{1}{e}$ via a greedy algorithm. The complete approximation algorithm for multiple sets is summarized in Algorithm 3.

**Lemma 2.7.** *For $n$ zero-mean normally distributed variables $X_1, X_2, \cdots, X_n$ and $\sigma_0 > 0$, let $f : 2^{[n]} \to \mathbb{R}$ be a set function such that for any $S \subseteq [n]$, $f(S) = \mathbb{E} \max_{i \in [n]} X_i$ where $X_i \sim \mathcal{N}(0, \sigma_0^2)$ for $i \in S$ and $X_i \equiv 0$ for $i \notin S$. Then $f$ is a submodular function.*

---

**Algorithm 3** An $O(\log n)$-approx algorithm distributing variances to variables in multiple sets

---

**Input:** Means $(\mu_1, \cdots, \mu_n)$, Sets $S_1, \cdots, S_m$
Initialize current best objective $M \leftarrow 0$ and candidate $(\hat{\sigma}_1, \cdots, \hat{\sigma}_n) \leftarrow (0, \cdots, 0)$;
Remove all sets with $|S_j| = 1$;
**for** $k \leftarrow 0$ to $\log_2 n$ **do**
    $(\sigma_1, \cdots, \sigma_n) \leftarrow (0, \cdots, 0)$;
    **while** # variables with variance $4^{-k} < \min(4^k, n)$ **do**
        Find the variable $i$ with $\sigma_i = 0$ such that after setting $\sigma_i^2 = 4^{-k}$, OBJ$(\boldsymbol{\sigma})$ is maximized;
        $\sigma_i^2 \leftarrow 4^{-k}$;
    **end while**
    **if** OBJ$(\boldsymbol{\sigma}) > M$ **then**
        $M \leftarrow$ OBJ$(\boldsymbol{\sigma})$;
        $\hat{\boldsymbol{\sigma}} \leftarrow \boldsymbol{\sigma}$;
    **end if**
**end for**
**Output:** $(\hat{\sigma}_1, \cdots, \hat{\sigma}_n)$

---

For CorrGraphVarAlloc that allows covariance, exactly the same algorithm gives a logarithmic approximation since the optimal objectives between the independent case GraphVarAlloc and the correlated case CorrGraphVarAlloc are bounded by a constant factor due to a constant correlation gap of the optimization problem.

**Lemma 2.8.** *For any Gaussian vector $(X_1, \cdots, X_n) \sim \mathcal{N}(\mu, \Sigma)$ with non-negative means, consider independent Gaussian variables $(Y_1, \cdots, Y_n) \sim \mathcal{N}(\mu, diag(\Sigma))$ with the same means and variances. Then*

$$\mathbb{E} \max(X_1, \cdots, X_n) \leq \frac{2e}{e-1} \mathbb{E} \max(Y_1, Y_2, \cdots, Y_n).$$

## Impact Statement

This paper presents work whose goal is to advance the field of Machine Learning. There are many potential societal consequences of our work, none which we feel must be specifically highlighted here.

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

# A. More Details on Variance Allocation in One Set

## A.1. Variance allocation for multiple independent variables

In this section, we propose several efficient approximation schemes that find near-optimal ways to distribute the variance across all random variables. We start with the simplest case VarAlloc, where there are $n$ normally distributed random variables $X_1 \sim \mathcal{N}(0, \sigma_1^2), X_2 \sim \mathcal{N}(0, \sigma_2^2), \cdots, X_n \sim \mathcal{N}(0, \sigma_n^2)$, and the to be distributed variances satisfy $\sigma_1^2 + \sigma_2^2 + \cdots + \sigma_n^2 = 1$.

Recall that OPT denotes the optimal objective of the following optimization problem:

$$\text{OPT} = \max_{\sigma_1, \cdots, \sigma_n} \mathbb{E} \max_{i \in [n]} X_i$$

$$\text{subject to} \quad \sum_{i=1}^n \sigma_i^2 = 1.$$

We have the following theorem.

**Theorem A.1.** *For any constant $\epsilon > 0$, there exists an algorithm with running time polynomial in $n$, which computes a variance vector $\boldsymbol{\sigma} = (\sigma_1, \cdots, \sigma_n)$ satisfying the optimization constraint, such that for $X_1 \sim \mathcal{N}(0, \sigma_1^2), X_2 \sim \mathcal{N}(0, \sigma_2^2), \cdots, X_n \sim \mathcal{N}(0, \sigma_n^2)$, $\mathbb{E} \max_{i \in [n]} X_i \geq \text{OPT} - \epsilon$.*

The main intuition for the PTAS is as follows. Firstly, all random variables with variance less than $\epsilon^2$ can be ignored, as their total contribution to the objective is negligible (i.e. $\tilde{O}(\epsilon)$). Secondly, the objective is Lipschitz with respect to each $\sigma_i$, thus we may only need to consider solutions with discrete $\sigma_i$ values. The above intuitions are formally described by Lemma 2.1 (main paper) and Lemma 2.2 (below).

**Lemma 2.2.** *Consider two vectors of standard deviations $(\sigma_1, \cdots, \sigma_n)$ and $(\sigma_1', \cdots, \sigma_n')$. Let $X_1, \cdots, X_n$ and $Y_1, \cdots, Y_n$ be two sets of mutually independent normal variables with $X_i \sim \mathcal{N}(\mu_i, \sigma_i^2)$ and $Y_i \sim \mathcal{N}(\mu_i, \sigma_i'^2)$, then*

$$\left| \mathbb{E} \max_{i \in [n]} X_i - \mathbb{E} \max_{i \in [n]} Y_i \right| \leq O(1) \cdot \sum_{i \in [n]} |\sigma_i - \sigma_i'|.$$

To prove Lemma 2.2, we prove the following much stronger lemma, and Lemma 2.2 can be viewed as a corollary by repeatedly applying Lemma A.2 to each index $i \in [n]$.

**Lemma A.2.** *Consider two normal variables $X \sim \mathcal{N}(\mu, \sigma_1^2)$ and $Y \sim \mathcal{N}(\mu, \sigma_2^2)$ with the same mean. For any random variable $T$ that is independent with $X$ and $Y$, let $X' = \max(X, T)$, and $Y' = \max(Y, T)$. Then*

$$|\mathbb{E} X' - \mathbb{E} Y'| \leq O(1) \cdot |\sigma_1 - \sigma_2|.$$

*Proof of Lemma A.2.* We can couple $X$ and $Y$ by assuming $X = \mu + \sigma_1 Z$ and $Y = \mu + \sigma_2 Z$, where $Z \sim \mathcal{N}(0, 1)$ is a standard normal variable, and $Z$ and $T$ are mutually independent. Then, we have:

$$
\begin{aligned}
|\mathbb{E} X' - \mathbb{E} Y'| &= |\mathbb{E} \max(X, T) - \mathbb{E} \max(Y, T)| \\
&\leq \mathbb{E} |\max(X, T) - \max(Y, T)| \\
&\leq \mathbb{E} |X - Y| \\
&= \mathbb{E} |\sigma_1 Z - \sigma_2 Z| \\
&= |\sigma_1 - \sigma_2| \mathbb{E} |Z|.
\end{aligned}
$$

Since $Z \sim \mathcal{N}(0, 1)$, its expected absolute value is $\mathbb{E}|Z| = \sqrt{\frac{2}{\pi}} = O(1)$. Thus, $|\mathbb{E} X' - \mathbb{E} Y'| \leq O(1) \cdot |\sigma_1 - \sigma_2|$, finishing the proof of the lemma.

$\square$

Now with the help of Lemma 2.1 and Lemma 2.2, we are ready to prove Theorem A.1.

*Proof of Theorem A.1.* Let k = $\frac{1}{\epsilon^2}$. We first show that we only need to distribute variance to at most $k$ variables.

When $n \geq k$, first observe that $\mathbb{E} \max_{i \in [n]} X_i$ and $\mathbb{E} \max(0, \max_{i \in [n]} X_i)$ are close (up to $O(2^{-1/\epsilon^2})$). This is true since $\Pr[\max_{i \in [n]} X_i < 0] \leq 2^{-k}$, and $\mathbb{E}[\max_{i \in [n]} X_i | \max_{i \in [n]} X_i < 0] \geq \mathbb{E}[X_1 | X_1 < 0] = -\sqrt{\frac{2}{\pi}} \sigma_1 = -O(1)$ for $\sigma_1 \leq 1$.

Then for any variance allocation $\sigma_1 \geq \sigma_2 \geq \cdots \geq \sigma_n$, we know that the objective value

$$\mathbb{E} \max(0, \max_{i \in [n]} X_i) \leq \mathbb{E} \max(0, \max_{i \in [k]} X_i) + \mathbb{E} \max(0, \max_{k+1 \leq i \leq n} X_i) = \mathbb{E} \max(0, \max_{i \in [k]} X_i) + O(\epsilon)$$

by Lemma 2.1[9]. Therefore, an algorithm that wants to obtain $O(\epsilon)$ loss does not need to distribute variance $< \epsilon^2$ to any variable. Also, as the total variance budget is 1, there can be at most $\frac{1}{\epsilon^2}$ variables with variance $> \epsilon^2$, thus $k \leq \frac{1}{\epsilon^2}$. This means that an algorithm that wants to obtain $O(\epsilon)$ loss only needs to distribute at least $\epsilon^2$ variance to at most $\frac{1}{\epsilon^2}$ variables.

By Lemma 2.2, if an algorithm can obtain the optimal objective conditioned on each $\sigma_i$ being an integer times $\epsilon^3$, the algorithm will have a loss at most $k\epsilon^3$, if at most $k$ variables have positive variance. Thus the following algorithm has $O(\epsilon)$ loss compared to the optimal objective:

---

**Algorithm 4** A PTAS algorithm distributing variances to zero-mean variables

---

    **Input:** $n \geq 0$
    $k \leftarrow \frac{1}{\epsilon^2}$;
    Set current maximum $M = 0$;
    **for** any $i > k$ **do**
        $\hat{\sigma}_i \leftarrow 0$
    **end for**
    **for** all possible $(\sigma_1, \cdots, \sigma_k)$, where $\sigma_i$ is an integral multiple of $\epsilon^3$ for every $i \in [k]$ **do**
        **if** $\sum_{i=1}^k \sigma_i^2 \leq 1$ and $\mathbb{E} \max_{i \in [k]} X_i > M$ **then**
            Set $(\hat{\sigma}_1, \cdots, \hat{\sigma}_k) \leftarrow (\sigma_1, \cdots, \sigma_k)$;
            $M \leftarrow \text{OBJ}(\boldsymbol{\sigma})$;
        **end if**
    **end for**
    **Output:** $(\hat{\sigma}_1, \cdots, \hat{\sigma}_n)$

---

Now we analyze the running time of the algorithm. There are at most $(\epsilon^{-3})^{1/\epsilon^2}$ different vectors of $\hat{\boldsymbol{\sigma}}$, and for each vector, we can efficiently compute the max of $k$ normal variables with given variance. Thus the algorithm is a polynomial time approximation scheme of the original objective.

$\square$

The algorithm can be easily extended to the case where the means of the variables are no longer 0.

**Theorem 1.1.** *Given an input to* VarAlloc *with* $\mu_1, \cdots, \mu_n \geq 0$ *and constant* $\epsilon > 0$*, there exists an algorithm with running time polynomial in* $n$*, that computes a variance vector* $(\hat{\sigma}_1, \cdots, \hat{\sigma}_n)$ *such that* $\mathbb{E} \max_{i \in [n]} X_i \geq \text{OPT} - \epsilon$.

*Proof of Theorem 1.1.* For any variance allocation $\sigma_1, \cdots, \sigma_n$, let $G_1 \cup G_2$ be the partition of all variables such that $G_1$ contains all variables $X_i$ with $\sigma_i > \epsilon^2$; $G_2$ contains all variables $X_i$ with $\sigma_i \leq \epsilon^2$. For each variable $X_i$, define $X_i' = X_i - \mu_i$.

---

    [9]For the clarity of presentation, we do not optimize the small terms.

Then $X_i' \sim \mathcal{N}(0, \sigma_i^2)$ is a zero-mean variable. We can bound the original objective as follows:

$$
\begin{aligned}
\mathbb{E}\max_{i \in [n]} X_i \quad &= \quad \mathbb{E}\max\left(\max_{X_i \in G_1} X_i, \max_{X_i \in G_2} X_i\right) \\[2mm]
&\leq \quad \mathbb{E}\max\left(\max_{X_i \in G_1} X_i, \max_{X_i \in G_2} \mu_i + \max\left(0, \max_{X_i \in G_2} X_i'\right)\right) \\[2mm]
&\leq \quad \mathbb{E}\max\left(\max_{X_i \in G_1} X_i, \max_{X_i \in G_2} \mu_i\right) + \mathbb{E}\max\left(0, \max_{X_i \in G_2} X_i'\right) \\[2mm]
&= \quad \mathbb{E}\max\left(\max_{X_i \in G_1} X_i, \max_{X_i \in G_2} \mu_i\right) + O(\epsilon).
\end{aligned}
$$

Here the first line is by splitting the objective into two groups of variables; the second line is by decomposing $X_i$ to $X_i' + \mu_i$, and upper-bounding $X_i'$ by $\max(0, X_i')$; the third line is by getting the positive term outside of the outer maximization; the last line is by using Lemma 2.1 to bound the maximum of multiple zero-mean variables. This means that to get $O(\epsilon)$ loss in the objective, it suffices to only distribute variance to $O(\frac{1}{\epsilon^2})$ variables, with each having variance at least $\epsilon^2$.

---

**Algorithm 1** A PTAS algorithm distributing variances to independent arbitrary-mean variables

1: **Input:** $\mu_1, \mu_2, \cdots, \mu_n \geq 0$
2: $k \leftarrow \frac{1}{\epsilon^2}$;
3: Set current maximum $M = 0$;
4: **for** all possible $k$ indices $1 \leq i_1 < i_2 < \cdots < i_k \leq n$ **do**
5:     **for** any $i \notin \{i_1, \cdots, i_k\}$ **do**
6:         Set $\hat{\sigma}_i = 0$;
7:     **end for**
8:     **for** all possible $(\sigma_{i_1}, \cdots, \sigma_{i_k})$, where $\sigma_{i_j}$ is an integral multiple of $\epsilon^3$ for every $j \in [k]$ **do**
9:         **if** $\sum_{j=1}^{k} \sigma_{i_j}^2 \leq 1$ **and** $\mathbb{E}\max_{i \in [n]} X_i > M$ **then**
10:           $(\hat{\sigma}_{i_1}, \cdots, \hat{\sigma}_{i_k}) \leftarrow (\sigma_{i_1}, \cdots, \sigma_{i_k})$;
11:           $M \leftarrow \mathrm{OBJ}(\boldsymbol{\sigma})$;
12:         **end if**
13:     **end for**
14: **end for**
15: **Output:** $\hat{\boldsymbol{\sigma}} = (\hat{\sigma}_1, \cdots, \hat{\sigma}_n)$

---

Similar to Theorem A.1, we can use a similar algorithm to enumerate all possible ways to distribute variances being multiple of $\epsilon^3$, see Algorithm 1 for details. The only difference from the zero-mean Algorithm 4 is that we now need to guess which $\frac{1}{\epsilon^2}$ variables should be distributed variances. The algorithm runs in polynomial time for any fixed $\epsilon > 0$, and obtains a solution with $O(\epsilon)$ loss compared to the optimal objective.

$\square$

### A.2. Variance distribution for correlated variables

We now proceed to the more general case, where there are $n$ jointly normally distributed random variables $(X_1, X_2, \cdots, X_n) \sim \mathcal{N}(\mu, \Sigma)$. We still assume that the total variance of the variables is bounded: $\sum_{i=1}^{n} \Sigma_{ii} \leq 1$, and we are also free to determine the positive semi-definite covariance matrix $\Sigma$ satisfying the constraint. Let OPT denote the optimal objective in this setting:

$$
\begin{aligned}
\mathrm{OPT} \quad &= \quad \max_{\Sigma} \mathbb{E}\max_{i \in [n]} X_i \\[2mm]
\text{subject to} \quad &\qquad \sum_{i=1}^{n} \Sigma_{ii} = 1.
\end{aligned}
$$

We have the following theorem.

**Theorem 1.2.** *Given an input to* CorrVarAlloc *and constant* $\epsilon > 0$, *there exists an algorithm with running time polynomial in* $n$, *that computes a covariance matrix* $\widehat{\Sigma}$ *such that for* $(X_1, \cdots, X_n) \sim \mathcal{N}(\mu, \widehat{\Sigma})$, $\mathbb{E} \max_{i \in [n]} X_i \geq \text{OPT} - \epsilon$.

The algorithm would be almost the same as Algorithm 1. Although we proved that all random variables with small variance $< \epsilon^2$ can be ignored in Lemma 2.1 for the correlated case, the Lemma that proves the smoothness of the objective with respect to the covariance is only shown for the independent case (Lemma 2.2). Therefore, we need to get a stronger smoothness lemma. We prove the following two lemmas: one shows that for any two joint distributions over $n$ variables, if they have a small Earth Mover's Distance, then their objective OBJ would be close; the other shows that for two Gaussian distributions with covariance matrices close element-wise, their Earth Mover's distance is small.

**Lemma 2.3.** *For two random vectors* $X = (X_1, X_2, \cdots, X_n)$ *and* $Y = (Y_1, Y_2, \cdots, Y_n)$ *drawn from distribution* $D_X$ *and* $D_Y$ *respectively, if the Earth Mover's Distance* $d_{EMD}(D_X, D_Y)$ *(defined by* $\ell_1$ *norm) between two distributions is at most* $\epsilon$, *then* $|\mathbb{E} \max_i X_i - \mathbb{E} \max_i Y_i| \leq \epsilon$.

*Proof of Lemma 2.3.* Consider the coupling $\gamma$ between $D_X$ and $D_Y$ that defines the Earth Mover's Distance

$$d_{EMD}(D_X, D_Y) = \mathbb{E}_{(X,Y) \sim \gamma} \|X - Y\|_1.$$

Notice that for any two vectors $X$ and $Y$, $|\max_i X_i - \max_i Y_i| \leq \|X - Y\|_1$. Thus

$$|\mathbb{E} \max_i X_i - \mathbb{E} \max_i Y_i| \leq \mathbb{E}_{(X,Y) \sim \gamma} |\max_i X_i - \max_i Y_i| \leq \mathbb{E}_{(X,Y) \sim \gamma} \|X - Y\|_1 = d_{EMD}(D_X, D_Y) \leq \epsilon.$$

$\square$

**Lemma 2.4.** *For any two dimension-$k$ Gaussian distributions with* $D_1 = \mathcal{N}(\mu, A)$ *and* $D_2 = \mathcal{N}(\mu, B)$, *suppose that covariance matrices* $A$ *and* $B$ *are close in entry-wise* $\ell_1$ *distance:* $\|A - B\|_{1,1} = \sum_{i,j} |A_{ij} - B_{ij}| \leq \epsilon$, *with* $\text{tr}(A) = \text{tr}(B) = 1$. *Then the Earth Mover Distance between* $D_1$ *and* $D_2$ *is upper bounded by* $\sqrt{k}\epsilon$.

*Proof of Lemma 2.4.* For any zero-mean multi-dimensional Gaussian distributions with Positive Semi-Definite (PSD) matrices $A$ and $B$, we upper bound the Earth Mover's Distance with their Wasserstein-2 ($W_2$) distance. The $W_2$ distance is defined similarly to the EMD distance, where EMD distance describes the smallest effort to transport between the two distributions using $\ell_1$ distance, while $W_2$ distance uses $\ell_2$ distance in the coupling. It is known that the EMD distance is always upper-bounded by $\sqrt{k}$ times the $W_2$ distance (via the Cauchy-Schwarz inequality $\|x\|_1 \leq \sqrt{k}\|x\|_2$):

$$d_{EMD}(D_1, D_2) \leq \sqrt{k} \cdot d_{W_2}(D_1, D_2),$$

and by (Dowson & Landau, 1982)

$$d_{W_2}^2(D_1, D_2) = \text{tr}\left(A + B - 2(A^{1/2}BA^{1/2})^{1/2}\right) = 2 - 2\text{tr}((A^{1/2}BA^{1/2})^{1/2}).$$

Now we bound $1 - \text{tr}((A^{1/2}BA^{1/2})^{1/2})$. By Fuchs-van de Graaf inequality (Fuchs & Van De Graaf, 2002),

$$1 - \text{tr}((A^{1/2}BA^{1/2})^{1/2}) \leq \frac{1}{2}\|A - B\|_*,$$

where $\|\cdot\|_*$ is the nuclear norm of a matrix. Since the nuclear norm of a matrix is upper bounded by its entry-wise $\ell_1$ norm (Johnston & Kribs, 2013), we have $\|A - B\|_* \leq \|A - B\|_{1,1} = \epsilon$. Thus

$$d_{EMD}(D_1, D_2) \leq \sqrt{k \cdot d_{W_2}^2(D_1, D_2)} = \sqrt{k} \cdot \sqrt{2 - 2\text{tr}((A^{1/2}BA^{1/2})^{1/2})} \leq \sqrt{k} \cdot \sqrt{2 \cdot \frac{1}{2}\epsilon} = \sqrt{k}\epsilon.$$

$\square$

Now we are ready to propose and analyze the algorithm for Theorem 1.2.

*Proof of Theorem 1.2.* The algorithm and its analysis are almost identical to Theorem 1.1, except we need a more fine-grained search due to the loss in the smoothness lemma (Lemma 2.4 vs. Lemma 2.2). Let $k = \frac{1}{\epsilon^2}$ and $\delta$ be to-be-determined parameters. When we search for the variance allocation, instead of searching for the variance vector with at most $k$ positive terms, we search for the covariance matrix of size at most $k^2$ for a subset of variables. As $|\Sigma_{ij}| \leq \sqrt{\Sigma_{ii}\Sigma_{jj}} \leq 1$, we only need to search for a covariance matrix with each element being an integral multiple of $\delta$ bounded between $-1$ and $1$.

The $\ell_1$ accuracy of the covariance matrix we get is $k^2\delta$. By Lemma 2.4, the Earth Mover's distance of the optimal solution and the matrix we get from the grid search algorithm is $O(\sqrt{k \cdot k^2\delta}) = O(\epsilon^{-3}\delta^{1/2})$. Thus, a grid size $\delta = \epsilon^8$ is enough to let the EMD distance be $O(\epsilon)$. See Algorithm 2 for the full algorithm.

---

**Algorithm 2** A PTAS algorithm distributing variances to arbitrary-mean correlated variables

> **Input:** $\mu_1, \mu_2, \cdots, \mu_n \geq 0$
> $k \leftarrow \frac{1}{\epsilon^2}$;
> Set current maximum $M = 0$;
> **for** all possible $k$ indices $1 \leq i_1 < i_2 < \cdots < i_k \leq n$ **do**
>     **for** any $i \notin \{i_1, \cdots, i_k\}$ **do**
>         **for** any $\ell \in [n]$ **do**
>             set $\widehat{\Sigma}_{i\ell} = \widehat{\Sigma}_{\ell i} = 0$;
>         **end for**
>     **end for**
>     **for** any $i, j \in \{i_1, \cdots, i_k\}$ **do**
>         **for** all possible values of $(\Sigma_{ij})$ being an integral multiple of $\epsilon^8$ with value in [-1,1] **do**
>             **if** $\Sigma$ is positive semi-definite, $\text{tr}(\Sigma) \leq 1$, **and** $\text{OBJ}(\Sigma) > M$ **then**
>                 $\widehat{\Sigma} \leftarrow \Sigma$;
>                 $M \leftarrow \text{OBJ}(\Sigma)$;
>             **end if**
>         **end for**
>     **end for**
> **end for**
> **Output:** $\widehat{\Sigma}$

---

$\square$

## B. More Details on Variance Allocation in Multiple Subsets

In this section, we study the multi-subset setting where there are $m > 1$ sets. Recall that each set's contribution is again defined by $\max_{i \in S_j} X_i$, and our objective is to maximize the total expected contribution of the $m$ sets. We assume that the $n$ variables $X_i \sim \mathcal{N}(\mu_i, \sigma_i^2)$ are independent and normally distributed, and we need to distribute a total variance budget 1. To be more precise, the optimal objective OPT is defined as

$$\text{OPT} = \max_{\sigma_1, \cdots, \sigma_n} \mathbb{E} \sum_{j=1}^{m} \max_{i \in S_j} X_i$$

$$\text{subject to} \quad \sum_{i=1}^{n} \sigma_i^2 = 1.$$

An equivalent way to define the instance is to use a bipartite graph. The characterization graph of the instance is a bipartite graph with $n$ vertices on one side representing variables $X_i$ for $i \in [n]$, and $m$ vertices on the other side representing sets $S_j$ for $j \in [n]$, with an edge $(i, j)$ exists if and only if $i \in S_j$.

When the sets are large in size, the PTAS algorithms in the previous sections are still applicable. Intuitively, this is true since OPT is of order $\Theta(m)$, and an additive loss $O(\epsilon)$ for each set leads to a total loss $O(m\epsilon)$. However, when the sets are relatively small, it is possible that the optimal variance allocation algorithm splits the total budget evenly among all

variables, resulting in each set having $o(\epsilon)$ contribution to the total objective. In this case, a grid search algorithm with grid points being small constants is not even sufficient to get a subpolynomial competitive ratio.

**Theorem 2.5.** *Suppose $m = n$, and $S_j = \{X_j, X_{j+1}\}$ for every $j \in [n]$ (assume that $X_{n+1} = X_1$). If all variables have the same mean, an optimal way to distribute variance is to set $\sigma_i^2 = \frac{1}{n}$ for each $i$.*

*Proof of Theorem 2.5.* For the special case where there are only two variables in each set, the optimization objective can be expressed by $\sigma$ in a closed form. Notice that the contribution of each set $S_j$ is

$$\mathbb{E}\max(X_j, X_{j+1}) = \mathbb{E}\frac{X_j + X_{j+1} + |X_j - X_{j+1}|}{2} = \mu + \frac{1}{2}\mathbb{E}|X_j - X_{j+1}|.$$

As $X_j \sim \mathcal{N}(\mu, \sigma_j^2)$ and $X_{j+1} \sim \mathcal{N}(\mu, \sigma_{j+1}^2)$, $|X_j - X_{j+1}|$ is the absolute value of a normally distributed variable with variance $\sigma_j^2 + \sigma_{j+1}^2$, and has mean $\sqrt{\frac{2}{\pi}} \cdot \sqrt{\sigma_j^2 + \sigma_{j+1}^2}$. Thus, the total objective can be written as

$$\sum_{j=1}^{n} \mathbb{E}\max(X_j, X_{j+1}) = \sum_{j=1}^{n}\left(\mu + \sqrt{\frac{1}{2\pi}} \cdot \sqrt{\sigma_j^2 + \sigma_{j+1}^2}\right)$$

$$= n\mu + \sqrt{\frac{1}{2\pi}}\sum_{j=1}^{n}\sqrt{\sigma_j^2 + \sigma_{j+1}^2}.$$

Thus it suffices to optimize the second term. By the Cauchy-Schwarz inequality,

$$\left(\sum_{j=1}^{n}\sqrt{\sigma_j^2 + \sigma_{j+1}^2}\right)^2 \leq \left(\sum_{j=1}^{n}(\sigma_j^2 + \sigma_{j+1}^2)\right)\left(\sum_{j=1}^{n}1^2\right)$$

$$= 2n.$$

The equality holds if and only if $\sigma_j^2 + \sigma_{j+1}^2 = \sigma_{j+1}^2 + \sigma_{j+2}^2 = \frac{2}{n}$ for every $j \in [n]$. Thus, setting all variables to have variance $\frac{1}{n}$ is an optimal solution to the original optimization problem.

$\square$

## B.1. Instance generated by a random graph

In this section, we study the setting where all variables are independent with mean $\mu_i = 0$ and total variance budget 1. The bipartite characterization graph generation follows Erdös–Rényi model with each edge appearing in the graph with probability $p$. In other words, for any variable $X_i$, $i \in S_j$ with probability $p$.

We observe that for a denser random graph (with larger $p$), fewer variables will have non-negligible variance in the optimal allocation. This is formally characterized in the following theorem.

**Theorem 1.6.** *For any $\delta > 0$ and a random instance of GraphVarAlloc with $n, m \to \infty$, with probability at least $1 - \delta$, there are $\Theta(\frac{1}{p})$ variables allocated variance $\Omega(p)$.*

*Proof of Theorem 1.6.* We first give an upper bound on the objective we can obtain in the random graph. Consider the following variance allocation: $\sigma_1^2 = \sigma_2^2 = \cdots = \sigma_{\frac{1}{p}}^2 = p$, $\sigma_{\frac{1}{p}+1} = \cdots = \sigma_n = 0$. Then for every set $S_j$, as any $i \in [n]$ appears in $S_j$ with probability $p$, the probability that $S_j$ contains at least one element in range $[1, \frac{1}{p}]$ is at least $1 - (1-p)^{1/p} > 1 - \frac{1}{e}$. Notice that when at least one $X_i$ in $S_j$ has variance $p$, the contribution of $S_j$ to the objective is at least $\mathbb{E}\max(X_1, 0) = \sqrt{\frac{p}{2\pi}} = \Omega(\sqrt{p})$. Thus, the optimal objective of the random graph is at least $\Omega(m\sqrt{p})$.

As the total variance budget of all variables is 1, there can be at most $O(\frac{1}{p})$ variables with large variance $\Omega(p)$. Now we show that if there are $o(\frac{1}{p})$ variables with variance $\Omega(p)$, then the total objective value is much smaller than the objective obtained by the allocation in the previous paragraph. To be more precise, we prove that for a small constant $\epsilon > 0$, if there are fewer than $\frac{\epsilon}{p}$ variables with variance at least $\epsilon p$, the total objective is only $o(m\sqrt{p})$.

For any variance vector $\sigma$, partition the variables into two groups $G_1$ and $G_2$. $G_1$ contains variables with variance at least $\epsilon p$, while $G_2$ contains all other variables with small variance. For any set $S_j$, let $s_j = \sum_{i \in S_j} \sigma_i^2$. The set's contribution to the total objective can be bounded by

$$
\begin{aligned}
\mathbb{E} \max_{i \in S_j} X_i &\leq \mathbb{E} \max \left( 0, \max_{i \in S_j \cap G_1} X_i, \max_{i \in S_j \cap G_2} X_i \right) \\
&\leq \mathbb{E} \sum_{i \in S_j \cap G_1} \max(0, X_i) + \mathbb{E} \max \left( 0, \max_{i \in S_j \cap G_2} X_i \right) \\
&= \mathbb{E} \sum_{i \in S_j \cap G_1} \max(0, X_i) + \sqrt{s_j} \mathbb{E} \max \left( 0, \max_{i \in S_j \cap G_2} \frac{X_i}{\sqrt{s_j}} \right) \\
&\leq \mathbb{E} \sum_{i \in S_j \cap G_1} \max(0, X_i) + \sqrt{s_j} \cdot O \left( \sqrt{\frac{\epsilon p}{s_j} \ln \frac{s_j}{\epsilon p}} \right) \\
&= \sqrt{\frac{1}{2\pi}} \sum_{i \in S_j \cap G_1} \sigma_i + \sqrt{\epsilon p} \cdot O \left( \sqrt{\ln \frac{s_j}{\epsilon p}} \right),
\end{aligned}
$$

here the last inequality is by Lemma 2.1. If we sum up the total contribution for all sets $S_j$, notice that with high probability every variable $X_i$ is only in $O(pm)$ sets, and $s_j = O(p)$. Thus with high probability

$$
\begin{aligned}
\mathbb{E} \sum_{j \in [m]} \max_{i \in S_j} X_i &\leq \sqrt{\frac{1}{2\pi}} \sum_{i \in G_1} \sigma_i \sum_{j : S_j \ni i} 1 + m \cdot O(\sqrt{\epsilon p \ln \frac{1}{\epsilon}}) \\
&= \sum_{i \in G_1} \sigma_i \cdot O(pm) + O(m \sqrt{\epsilon p \ln \frac{1}{\epsilon}}) \\
&\leq O(pm) \sqrt{\left( \sum_{i \in G_1} \sigma_i^2 \right)} \cdot \sqrt{\left( \sum_{i \in G_1} 1^2 \right)} + O(m \sqrt{\epsilon p \ln \frac{1}{\epsilon}}) \\
&\leq O(pm) \cdot 1 \cdot \sqrt{\frac{\epsilon}{p}} + O(m \sqrt{\epsilon p \ln \frac{1}{\epsilon}}) \\
&= O(m \sqrt{p\epsilon \ln \frac{1}{\epsilon}}).
\end{aligned}
$$

This means that such an allocation only leads to $O(\sqrt{\epsilon \ln \frac{1}{\epsilon}})$ fraction of the optimal objective (which is $O(m\sqrt{p})$), thus is negligible when $\epsilon$ is small.

$\square$

### B.2. Instance generated by arbitrary graph

In this section, we provide an efficient approximation algorithm for GraphVarAlloc with a competitive ratio $O(\log n)$.

**Theorem 1.3.** *Given an input to* GraphVarAlloc*, there exists an algorithm with a running time polynomial in $n$ that computes a variance vector $(\hat{\sigma}_1, \cdots, \hat{\sigma}_n)$ such that $\mathbb{E} \sum_{j=1}^{m} \max_{i \in S_j} X_i \geq \Omega \left( \frac{1}{\log n} \right)$ OPT.*

*Proof of Theorem 1.3.* We will iteratively simplify the problem and reduce the search space of the optimal variance vector, each time only losing a small approximation factor.

Firstly, we can remove all sets $S_j$ with size 1. This is because no matter what variance we allocate to the variable $X_i$ in the set, its contribution to the total objective $\mathbb{E} \max_{i \in S_j} X_i = \mu_i$ does not change. Therefore, if after removing all singleton sets we get a competitive ratio $\alpha$ to the new instance, the algorithm gets $\alpha$ competitive ratio to the original instance.

Next, we will transform our problem to remove the dependency on $\mu_i$ and only lose a constant factor in the competitive ratio. For each $i \in [n]$, let $X_i' \sim \mathcal{N}(0, \sigma_i^2)$ be a zero-mean normal variable with the same variance as $X_i$. For any set $S_j$, by Lemma B.1 we have

$$\mathbb{E} \max_{i \in S_j} X_i = \mathbb{E} \max_{i \in S_j}(\mu_i + X_i') \leq \mathbb{E} \max_{i \in S_j} \mu_i + \mathbb{E} \max_{i \in S_j} \max(0, X_i') \leq \mathbb{E} \max_{i \in S_j} \mu_i + 2\mathbb{E} \max_{i \in S_j} X_i'.$$

Notice that $\mathbb{E} \max_{i \in S_j} \mu_i$ is an obtainable objective via setting $\sigma_i = 0$ for every variable $X_i$; for any $\sigma$ vector that maximizes the zero-mean objective $\mathbb{E} \max_{i \in S_j} X_i'$, using the same variance allocation only leads to a larger objective in the original instance with $\mu_i \geq 0$. Thus solving the problem with each variable having zero mean leads to a 3-approximation of the original problem.

Next, we will bound the search range of $\sigma$. By Lemma 2.6, considering $\sigma_i^2$ being powers of 4 only loses a factor of 2 compared to the optimal solution. Thus we will only search for $\sigma$ with each $\sigma_i^2 = 4^{-k}$ or 0 for some integer $k$.

For any such $\sigma$, partition the variables $X_1, \cdots, X_n$ to the following $O(\log n)$ groups. Let $K = \log_2 n$. For each $k \leq K$, let $G_k$ contains all variables with variance $4^{-k}$. Let $G_{K+1}$ contains all other variables with variance smaller than $\frac{1}{n^2}$. Then for any set $S_j$,

$$\mathbb{E} \sum_j \max_{i \in S_j} X_i = \mathbb{E} \sum_j \max\left(\max_{X_i \in S_j \cap G_0} X_i, \max_{X_i \in S_j \cap G_1} X_i, \cdots, \max_{X_i \in S_j \cap G_{K+1}} X_i\right)$$

$$\leq \sum_{k=0}^{K} \sum_j \mathbb{E} \max(0, \max_{X_i \in S_j \cap G_k} X_i) + \sum_j \mathbb{E} \max(0, \max_{X_i \in S_j \cap G_{K+1}} X_i)$$

$$\leq \sum_{k=0}^{K} \sum_j \mathbb{E} \max(0, \max_{X_i \in S_j \cap G_k} X_i) + \sum_j O\left(\frac{1}{n}\sqrt{\ln n}\right), \tag{17}$$

here the last inequality is by Lemma 2.1. When setting all variables to have variance $\frac{1}{n}$, for each set $S_j$, $\mathbb{E} \max_{i \in S_j} X_i = \Omega(\frac{1}{\sqrt{n}}) = \omega\left(\frac{1}{n}\sqrt{\ln n}\right)$, thus the contribution from variables in group $G_{K+1}$ in (17) is negligible and we can ignore them. Notice that when setting all variables to have variance either $4^{-k}$ or 0, the optimal objective is at least $\sum_j \mathbb{E} \max(0, \max_{X_i \in S_j \cap G_k} X_i)$ from this specific $\sigma$.[10] Thus by solving the problem where all variables have the same variance $2^{-k}$, the obtained objective is at least $O(\frac{1}{\log n})$ fraction of the optimal objective for one of $k \leq \log_2 n$.

---

**Algorithm 3** An $O(\log n)$-approx algorithm distributing variances to variables in multiple sets

---

**Input:** Means $(\mu_1, \cdots, \mu_n)$, Sets $S_1, \cdots, S_m$
Initialize current best objective $M \leftarrow 0$ and candidate $(\hat{\sigma}_1, \cdots, \hat{\sigma}_n) \leftarrow (0, \cdots, 0)$;
Remove all sets with $|S_j| = 1$;
**for** $k \leftarrow 0$ to $\log_2 n$ **do**
    $(\sigma_1, \cdots, \sigma_n) \leftarrow (0, \cdots, 0)$;
    **while** # variables with variance $4^{-k} < \min(4^k, n)$ **do**
        Find the variable $i$ with $\sigma_i = 0$ such that after setting $\sigma_i^2 = 4^{-k}$, $\text{OBJ}(\sigma)$ is maximized;
        $\sigma_i^2 \leftarrow 4^{-k}$;
    **end while**
    **if** $\text{OBJ}(\sigma) > M$ **then**
        $M \leftarrow \text{OBJ}(\sigma)$;
        $\hat{\sigma} \leftarrow \sigma$;
    **end if**
**end for**
**Output:** $(\hat{\sigma}_1, \cdots, \hat{\sigma}_n)$

---

Now we want to solve the following problem: find the set $S \subseteq [n]$ of variables with $|S| \leq 4^k$, such that when all variables $X_i \in S$ have variance $\sigma_i^2 = 4^{-k}$ and all variables not in $S$ have variance 0, the total objective is maximized. By Lemma 2.7

---

[10] For every set $S_j$, if $S_j$ contains variables from only 1 group, the objective we get is actually $\mathbb{E} \max_{X_i \in S_j \cap G_k} X_i$ instead of $\mathbb{E} \max(0, \max_{X_i \in S_j \cap G_k} X_i)$.

this is a submodular maximization problem with a cardinality constraint, and a greedy solution obtains $1 - \frac{1}{e}$ fraction of the optimal solution (Nemhauser et al., 1978). The complete $O(\log n)$-approx algorithm is specified in Algorithm 3.

$\square$

**Lemma B.1.** *For $n \geq 2$ independent normally distributed variables $X_1, X_2, \cdots, X_n$ with non-negative means $\mu_1, \cdots, \mu_n \geq 0$,*

$$\mathbb{E} \max(X_1, \cdots, X_n) \geq (1 - 2^{1-n})\mathbb{E} \max(0, X_1, \cdots, X_n).$$

*Proof.* We consider the following way to generate each $X_i \sim \mathcal{N}(\mu_i, \sigma_i^2)$. Firstly, a positive value $Y_i$ is generated from a half-normal distribution with variance $\sigma_i^2$. In other words, $Y_i$ is the absolute value of a normal variable with mean 0 and variance $\sigma_i^2$. Next, a sign $Z_i$ is generated uniformly from $\{-1, 1\}$, and $X_i$ is determined by $\mu_i + Y_i Z_i$. For any fixed $Y_1, \cdots, Y_n$ we show that

$$\mathbb{E}_Z \max(\mu_1 + Y_1 Z_1, \cdots, \mu_n + Y_n Z_n) \geq \mathbb{E}_Z \max(0, \mu_1 + Y_1 Z_1, \cdots, \mu_n + Y_n Z_n).$$

Suppose that $k = \arg \max_i(\mu_i + Y_i)$. Notice that when $Z_k = 1$, which happens with probability $\frac{1}{2}$, $\max(\mu_i + Y_i Z_i) = \max(0, \max(\mu_i + Y_i Z_i)) = \mu_k + Y_k$. When $Z_1 = Z_2 = \cdots = Z_n = -1$, which happens with probability $\frac{1}{2^n}$, let $t = \max(\mu_i + Y_i Z_i)$, then $t \geq -\mu_k - Y_k$. In other cases, $\max(\mu_i + Y_i Z_i) = \max(0, \max(\mu_i + Y_i Z_i)) \geq 0$, and let $W \geq 0$ be the conditional expectation of $\max(\mu_i + Y_i Z_i)$ in this case. Then we have

$$\mathbb{E} \max(\mu_i + Y_i Z_i) = \frac{1}{2}(\mu_k + Y_k) + \frac{1}{2^n} t + \left(\frac{1}{2} - \frac{1}{2^n}\right) W,$$

and

$$\mathbb{E} \max(0, \max(\mu_i + Y_i Z_i)) = \frac{1}{2}(\mu_k + Y_k) + \frac{1}{2^n} \cdot \max(0, t) + \left(\frac{1}{2} - \frac{1}{2^n}\right) W.$$

As $t \geq -\mu_k - Y_k$,

$$\frac{\mathbb{E} \max(0, \max(\mu_i + Y_i Z_i))}{\mathbb{E} \max(\mu_i + Y_i Z_i)} \leq \frac{\frac{1}{2}}{\frac{1}{2} - \frac{1}{2^n}} = \frac{2^{n-1}}{2^{n-1} - 1}$$

for $n \geq 2$. Then the original lemma holds by taking the expectation of the above inequality over all possible $Y$. $\square$

**Lemma 2.6.** *For $n$ zero-mean normally distributed variables $X_1, X_2, \cdots, X_n$ with standard deviation $\sigma_1, \cdots, \sigma_n$, suppose that there are $n$ other zero-mean normally distributed variables $X_1', X_2', \cdots, X_n'$ with standard deviation $\sigma_1', \cdots, \sigma_n'$ satisfying $\sigma_i \leq \sigma_i' \leq 2\sigma_i$, then $\mathbb{E} \max X_i \leq \mathbb{E} \max X_i' \leq 2\mathbb{E} \max X_i$.*

*Proof of Lemma 2.6.* For each $i \in [n]$, we couple the generation of $X_i$ and $X_i'$ as follows. Firstly, a positive value $Y_i$ is generated from a half-normal distribution with variance $\sigma_i^2$. Next, a sign $Z_i$ is generated uniformly from $\{-1, 1\}$. Then let $X_i = Y_i Z_i$, and $X_i' = \alpha_i Y_i Z_i$ where $\alpha_i = \frac{\sigma_i'}{\sigma_i}$. Denote $Y_i' = \alpha_i Y_i$. As $X_i'$ has the same distribution as $\alpha_i X_i$, the above procedure correctly generates $X_i \sim \mathcal{N}(0, \sigma_i^2)$ and $X_i' \sim \mathcal{N}(0, (\sigma_i')^2)$.

For fixed $Y = (Y_1, \cdots, Y_n)$, we study the relationship between $\mathbb{E} \max X_i$ and $\mathbb{E} \max X_i'$. Let $Y^{(k)}$ denote the $k$th largest element in $Y$. Notice that when the largest element in $Y$ has sign $Z_i = 1$, $\max_i X_i = Y^{(1)}$, and this happens with probability $\frac{1}{2}$. When $Y^{(1)}$ has sign $-1$ and $Y^{(2)}$ has sign $+1$, $\max_i X_i = Y^{(2)}$ and this happens with probability $\frac{1}{4}$. Similarly, for every $k \leq n$, $\max_i X_i = Y^{(k)}$ with probability $2^{-k}$. When all $Z_i$ are $-1$, $\max_i X_i = -Y^{(n)}$, and this happens with probability $2^{-n}$. Thus

$$\mathbb{E} \max_i X_i = \sum_{k=1}^{n} 2^{-k} Y^{(k)} + 2^{-n}(-Y^{(n)}) = \sum_{k=1}^{n-1} 2^{-k} Y^{(k)}.$$

Similarly,

$$\mathbb{E} \max_i X_i' = \sum_{k=1}^{n-1} 2^{-k} Y'^{(k)}.$$

For $Y = (Y_1, \cdots, Y_n)$ and $Y' = (\alpha_1 Y_1, \alpha_2 Y_2, \cdots, \alpha_n Y_n)$ with $1 \le \alpha_i \le 2$ for every $i \in [n]$, $Y^{(k)} \le Y'^{(k)} \le 2Y^{(k)}$ for every $k \in [n]$. Thus for any fixed $Y$,

$$\mathbb{E} \max_i X_i = \sum_{k=1}^{n-1} 2^{-k} Y^{(k)} \le \mathbb{E} \max_i X_i' = \sum_{k=1}^{n-1} 2^{-k} Y'^{(k)} \le 2\mathbb{E} \max_i X_i.$$

The original theorem holds by taking the expectation of the above inequality over all possible $Y$. $\qquad \square$

**Lemma 2.7.** *For $n$ zero-mean normally distributed variables $X_1, X_2, \cdots, X_n$ and $\sigma_0 > 0$, let $f : 2^{[n]} \to \mathbb{R}$ be a set function such that for any $S \subseteq [n]$, $f(S) = \mathbb{E} \max_{i \in [n]} X_i$ where $X_i \sim \mathcal{N}(0, \sigma_0^2)$ for $i \in S$ and $X_i \equiv 0$ for $i \notin S$. Then $f$ is a submodular function.*

*Proof of Lemma 2.7.* It suffices to prove the lemma for $\sigma_0 = 1$, as the expected maximum with $X_i \sim \mathcal{N}(0, \sigma_0^2)$ is just $\sigma_0$ fraction of the expected maximum with $X_i \sim \mathcal{N}(0, 1)$.

For any $k \in [n]$, let $g(k) = \mathbb{E} \max(0, X_1, X_2, \cdots, X_k)$. Then for any $S$ with $|S| = k$, $f(S) = g(k)$, except when $|S| = n$, $g(n) = \mathbb{E} \max(0, X_1, X_2, \cdots, X_n) > \mathbb{E} \max(X_1, X_2, \cdots, X_n) = f(S)$. Thus to prove $f(S)$ is submodular, we only need to prove the submodularity of $g$, or equivalently for any $k \ge 1$, $g(k+1) - g(k) < g(k) - g(k-1)$.

Let $\Phi$ be the CDF of the standard normal distribution $\mathcal{N}(0, 1)$. Then $\max(X_1, \cdots, X_k)$ has CDF $\Phi^k$, and

$$g(k) = \mathbb{E} \max(0, X_1, X_2, \cdots, X_k) = \int_0^\infty (1 - \Phi^k(x))dx.$$

Then for any $k$,

$$
\begin{aligned}
g(k) - g(k-1) &= \int_0^\infty (1 - \Phi^k(x))dx - \int_0^\infty (1 - \Phi^{k-1}(x))dx \\
&= \int_0^\infty \Phi^{k-1}(x)(1 - \Phi(x))dx \\
&\ge \int_0^\infty \Phi^k(x)(1 - \Phi(x))dx \\
&= \int_0^\infty (1 - \Phi^{k+1}(x))dx - \int_0^\infty (1 - \Phi^k(x))dx \\
&= g(k+1) - g(k).
\end{aligned}
$$

Thus $g$ is a submodular function, which implies that $f$ is a submodular set function.

$\qquad \square$

**Lemma B.2.** *For any $a, b, c \in \mathbb{R}$,*
$$\max(a, b) + \max(a, c) \ge \max(a, b, c) + a.$$

*Proof of Lemma B.2.* As $b$ and $c$ are symmetric in the desired inequality, without loss of generality assume $b \ge c$. Discuss the order of the variables as follows.

- When $a \ge b \ge c$: $LHS = a + a \ge a + a = RHS$.

- When $b \ge a \ge c$: $LHS = b + a \ge b + a = RHS$.

- When $b \ge c \ge a$: $LHS = b + c \ge b + a = RHS$.

Thus the lemma holds for all possible orderings of $a, b, c$. $\qquad \square$

**Lemma B.3.** *For any $a, b, c, d \in \mathbb{R}$,*

$$
\begin{aligned}
&3\max(a, b, c, d) + \max(a, d) + \max(b, d) + \max(c, d) \\
\le\ & 2\max(a, b, d) + 2\max(a, c, d) + 2\max(b, c, d).
\end{aligned}
$$

*Proof of Lemma B.3.* By applying Lemma B.2 to $\max(a,d)$, $b$ and $c$, we get

$$\max(a,b,d) + \max(a,c,d) \geq \max(a,b,c,d) + \max(a,d).$$

In the same way, we can get

$$\max(a,b,d) + \max(b,c,d) \geq \max(a,b,c,d) + \max(b,d),$$

$$\max(a,c,d) + \max(b,c,d) \geq \max(a,b,c,d) + \max(c,d).$$

The lemma holds by adding the above three inequalities. $\qquad\square$

**Theorem 1.5.** *Let $I_k$ be an instance for* GraphVarAlloc *with all $\binom{n}{k}$ sets of size $k$: for every $1 \leq i_1 < i_2 < \cdots < i_k \leq n$, there is a set $S_j = \{i_1, \cdots, i_k\}$ in this instance. For any fixed variance allocation $\boldsymbol{\sigma}$, let $f_{\boldsymbol{\sigma}}(k) = \frac{1}{\binom{n}{k}}\mathrm{OBJ}_{I_k}(\boldsymbol{\sigma})$ be the expected per-set contribution. Then $f_{\boldsymbol{\sigma}}(k)$ is a concave function with respect to $k$.*

*Proof of Theorem 1.5.* For any fixed $\boldsymbol{\sigma}$, for simplicity let $g = f_{\boldsymbol{\sigma}}$. To prove the concavity of $g$, we prove that for any $k \geq 3$,

$$g(k) + g(k-2) \leq 2g(k-1). \tag{18}$$

Notice that $g(k)$ is the per-set contribution to the objective. We can get an equivalent definition: randomly sample $k$ elements $(i_1, \cdots, i_k)$ from $[n]$ without replacement, then

$$g(k) = \mathbb{E}_{(X_1,\cdots,X_n)}\mathbb{E}_{(i_1,\cdots,i_k)}\max(X_{i_1}, X_{i_2}, X_{i_3}, X_{i_4}\cdots, X_{i_k}).$$

Such a sampling can be viewed as first sample $k-3$ elements $(i_4, \cdots, i_k)$ without replacement, then sample 3 more elements $(i_1, i_2, i_3)$ without replacement. $g(k-1)$ can be viewed as first sample $k-3$ elements $(i_4, \cdots, i_k)$ without replacement, then sample 3 more elements $(i_1, i_2, i_3)$ without replacement, then choose two variables from $X_{i_1}$, $X_{i_2}$ and $X_{i_3}$:

$$
\begin{aligned}
g(k-1) \;=\; & \mathbb{E}_{(X_1,\cdots,X_n)}\mathbb{E}_{(i_1,\cdots,i_k)}\frac{1}{3}\bigg( \max(X_{i_1}, X_{i_2}, X_{i_4}\cdots, X_{i_k}) \\
& + \max(X_{i_1}, X_{i_3}, X_{i_4}\cdots, X_{i_k}) \\
& + \max(X_{i_2}, X_{i_3}, X_{i_4}, \cdots, X_{i_k})\bigg).
\end{aligned}
$$

$g(k-2)$ can be viewed as first sample $k-3$ elements $(i_4, \cdots, i_k)$ without replacement, then sample 3 more elements $(i_1, i_2, i_3)$ without replacement, then choose one variable from $X_{i_1}$, $X_{i_2}$ and $X_{i_3}$:

$$
\begin{aligned}
g(k-2) \;=\; & \mathbb{E}_{(X_1,\cdots,X_n)}\mathbb{E}_{(i_1,\cdots,i_k)}\frac{1}{3}\bigg( \max(X_{i_1}, X_{i_4}, \cdots, X_{i_k}) \\
& + \max(X_{i_2}, X_{i_4}, \cdots, X_{i_k}) \\
& + \max(X_{i_3}, X_{i_4}, \cdots, X_{i_k})\bigg).
\end{aligned}
$$

To prove that $g(k) + g(k-2) \leq 2g(k-1)$, it suffices to show that this inequality holds even for every realization of $(X_1, \cdots, X_n)$ and $(i_1, \cdots, i_k)$. After removing the expectation quantifier, we get

$$g(k) = \max(X_{i_1}, X_{i_2}, X_{i_3}, X_{i_4}, \cdots, X_{i_k}),$$

$$
\begin{aligned}
g(k-1) \;=\; & \frac{1}{3}\bigg( \max(X_{i_1}, X_{i_2}, X_{i_4}, \cdots, X_{i_k}) + \max(X_{i_1}, X_{i_3}, X_{i_4}, \cdots, X_{i_k}) \\
& + \max(X_{i_2}, X_{i_3}, X_{i_4}, \cdots, X_{i_k})\bigg),
\end{aligned}
$$

$$g(k-2) \quad = \quad \frac{1}{3}\bigg( \max(X_{i_1}, X_{i_4}, \cdots, X_{i_k}) + \max(X_{i_2}, X_{i_4}, \cdots, X_{i_k})$$

$$+ \max(X_{i_3}, X_{i_4}, \cdots, X_{i_k}) \bigg).$$

By applying $a = X_{i_1}$, $b = X_{i_2}$, $c = X_{i_3}$ and $d = \max(X_{i_4}, \cdots, X_{i_k})$ to Lemma B.3, we exactly get $3g(k) + 3g(k-2) \leq 6g(k-1)$. By taking the expectation over all possible realizations of $(X_1, \cdots, X_n)$ and $(i_1, \cdots, i_k)$ we have finished the proof of (18), thus the concavity of the function.

□

**Lemma 2.8.** *For any Gaussian vector $(X_1, \cdots, X_n) \sim \mathcal{N}(\mu, \Sigma)$ with non-negative means, consider independent Gaussian variables $(Y_1, \cdots, Y_n) \sim \mathcal{N}(\mu, diag(\Sigma))$ with the same means and variances. Then*

$$\mathbb{E} \max(X_1, \cdots, X_n) \leq \frac{2e}{e-1} \mathbb{E} \max(Y_1, Y_2, \cdots, Y_n).$$

*Proof of Lemma 2.8.* For any $t \geq 0$, we analyze $\Pr[\max_i X_i \geq t]$ and $\Pr[\max_i Y_i \geq t]$. We first prove that

$$\Pr[\max_i Y_i \geq t] \geq \frac{e-1}{e} \Pr[\max_i X_i \geq t]. \tag{19}$$

For any $i \in [n]$, let $q_i = \Pr[Y_i \geq t]$. Notice that the marginal distributions $X_i$ and $Y_i$ are exactly the same normal distributions with identical mean $\mu_i$ and variance $\Sigma_{ii}$, we have $\Pr[X_i \geq t]$ is also $q_i$. Then for correlated variables $X_1, \cdots, X_n$, $\Pr[\max_i X_i \geq t] \leq \sum_i q_i$; for independent variables $Y_1, \cdots, Y_n$,

$$\Pr[\max_i Y_i \geq t] = 1 - \prod_i (1 - q_i) \geq 1 - \prod_i e^{-q_i} = 1 - e^{-\sum_i q_i}.$$

Let $q = \sum_i q_i$. When $q \geq 1$,

$$\frac{\Pr[\max_i Y_i \geq t]}{\Pr[\max_i X_i \geq t]} \geq \Pr[\max_i Y_i \geq t] \geq 1 - e^{-q} \geq 1 - e^{-1}.$$

When $q < 1$,

$$\frac{\Pr[\max_i Y_i \geq t]}{\Pr[\max_i X_i \geq t]} \geq \frac{1 - e^{-q}}{q}.$$

Let $g(q) = \frac{1-e^{-q}}{q}$. As $g'(q) = \frac{e^{-q}q - 1 + e^{-q}}{q^2} < 0$ for $0 < q < 1$, we have $g$ is a decreasing function on $(0, 1)$. Thus $g(q) > g(1) = 1 - e^{-1}$ for $0 < q < 1$. This finishes the proof of (19) for every $t \geq 0$.

Now we are ready to prove the original lemma using (19). Actually,

$$\begin{aligned}
\mathbb{E} \max(X_1, \cdots, X_n) \quad &\leq \quad \mathbb{E} \max(0, X_1, \cdots, X_n) \\
&= \quad \int_0^\infty \Pr[\max_i X_i \geq t] dt \\
&\leq \quad \int_0^\infty \frac{e}{e-1} \Pr[\max_i Y_i \geq t] dt \\
&= \quad \frac{e}{e-1} \mathbb{E} \max(0, Y_1, \cdots, Y_n) \\
&\leq \quad \frac{2e}{e-1} \mathbb{E} \max(Y_1, \cdots, Y_n).
\end{aligned}$$

Here the last inequality is from Lemma B.1 for $n \geq 2$. This finishes the proof of the lemma.

□

