# OpenReview forum: "Allocating Variance to Maximize Expectation"
_ICML.cc/2026/Conference — ICML 2026 regular_

### Official Review · Reviewer_6vyU · 2026-03-04

**Soundness:** 4
**Presentation:** 3
**Significance:** 4
**Originality:** 3
**Overall Recommendation:** 5
**Confidence:** 3

**Summary:**

This paper studies a stochastic optimization problem centered on maximizing the expectation of the supremum of Gaussian random variables under a global variance budget.
The main problem presented by this article is how to optimally allocate total variance across Gaussian variables (independent or correlated) in order to maximize either a single expected maximum or a sum of expected maxima over structured subsets.

The authors formalize four variants:
VarAlloc, CorrVarAlloc, GraphVarAlloc, and CorrGraphVarAlloc, progressively incorporating correlation and combinatorial structure via subsets.
For the single-set setting, they design additive polynomial-time approximation schemes (PTAS), while for the multiple-set (graph) setting, they provide an $O(\log n)$-approximation algorithm.

**Compliance With Llm Reviewing Policy:**

Affirmed.

**Final Justification:**

The rebuttal addressed my main concerns.

**Key Questions For Authors:**

Do you conjecture that the additive PTAS for VarAlloc and CorrVarAlloc is essentially optimal under standard complexity assumptions?

Do the structural and approximation results extend to sub-Gaussian or log-concave distributions?

The concentration phenomenon (variance supported on $\Theta(1 / p)$ variables) is intriguing.
Is there a more explicit characterization of optimal allocations for canonical graph families (e.g., cliques, cycles)?

**Limitations:**

Yes.

**Strengths And Weaknesses:**

Pros

Lemma 2.1 is a key technical contribution.
The improvement from is nontrivial and enables a true PTAS rather than a quasi-polynomial approximation scheme.

The PTAS for the single-set setting is technically interesting, especially the reduction in effective dimensionality via variance thresholding.
For the multi-set case, the reduction to submodular maximization and the resulting approximation is elegant and leverages classical results effectively.

The connection to auction revenue maximization and SGD dynamics provides helpful intuition and potential relevance to machine learning practice.

Cons

The paper states that tightness of the approximation guarantees is not established.
It remains unclear whether the PTAS and logarithmic approximations are optimal, or whether improved guarantees are possible in structured graph classes.

---

> ### Author Rebuttal · Authors · 2026-03-29
>
> We thank the reviewer for the thorough review and your strong support on finding the structural lemmas being nontrivial and the algorithmic reductions being elegant. We address your specific questions regarding optimality and extensions as follows.
>
> **1. Do you conjecture that the additive PTAS is optimal?**
> Yes, we strongly conjecture that it is. For a given covariance matrix, the exact evaluation of max of Gaussian can be already computationally hard. So it's  unlikely that the optimization problem itself can be easier to solved exactly than verifying a single instance of the max of a Gaussian.
>
> **2. Do the results extend to sub-Gaussian or log-concave distributions?**
> This is a great question. While the chaining arguments usually generalize to sub-Gaussian distributions (not sure about the log-concave distributions), the details need to be verified carefully and we would propose this as an important non-trivial direction for future work.
>
> **3. Explicit characterization for canonical graph families:**
> When each set has size 2, the cycles and cliques are well-defined and Theorem 2.5 has the cycle result (with every variable having the same variance for an instance with equal mean). For cliques, the uniform variance allocation is still optimal and the proof can be similar. For sets with size larger than 2, the optimization of the hypergraph instance can be much harder, and we will leave this as an open direction.

---

> > ### Author Rebuttal · Reviewer_6vyU · 2026-04-01
> >
> > My concerns have been adequately addressed.

---

### Official Review · Reviewer_vkB2 · 2026-03-08

**Soundness:** 2
**Presentation:** 2
**Significance:** 2
**Originality:** 3
**Overall Recommendation:** 3
**Confidence:** 4

**Summary:**

Overall, the research investigates a relevant issue. The main problem presented by this article is how to allocate a fixed variance budget, or more generally a covariance matrix with fixed trace, in order to maximize the expected maximum of Gaussian variables, either for a single set or for multiple subsets. The paper studies four formulations: VarAlloc, CorrVarAlloc, GraphVarAlloc, and CorrGraphVarAlloc. The claimed results are an additive PTAS for the single-set independent case, an additive PTAS for the single-set correlated case, O(log n) approximation algorithms for the multi-set independent and correlated cases, and structural statements about concavity of the per-set objective and concentration of the optimal allocation as the underlying set size increases. The paper also gives a stylized motivation through auctions and SGD / Ornstein-Uhlenbeck dynamics.

**Compliance With Llm Reviewing Policy:**

Affirmed.

**Key Questions For Authors:**

1. Please clarify the exact scope of Theorem 1.2. Is CorrVarAlloc handled for arbitrary means mu, or only for the zero-mean case? Right now the problem formulation, theorem statement, algorithm description, and appendix proof do not seem consistent.

2. In Algorithms 1 and 2, where are the feasibility constraints enforced? The pseudocode should explicitly require sum_i sigma_i^2 = 1 in the independent case and trace(Sigma) = 1 in the correlated case. Without these constraints, the algorithm descriptions are incomplete.

3. On page 2, the lower bound used to justify the additive PTAS claim seems incorrect. Could the authors check this constant and revise the discussion? More generally, I encourage the authors to carefully re-check constants and normalization throughout the paper.

4. How is OBJ evaluated during the grid search to the accuracy needed for the PTAS guarantee? The paper states polynomial-time algorithms in n for fixed epsilon, but it is not fully clear what computational model is assumed for evaluating expectations of maxima of Gaussian vectors, especially in the correlated case.

5. Can the authors strengthen the practical side of the paper? Right now the figures are useful intuition, but they do not validate the actual algorithms in realistic ML or auction settings. Even a modest empirical study of the proposed approximation algorithms would improve the submission significantly.

6. In Theorem 1.6 and the random-graph discussion, what assumptions on p are intended in the asymptotic regime? It would help to state clearly whether p is a constant, whether it may depend on n, and what scaling is required for the concentration statement.

**Limitations:**

No.

The paper includes a brief impact statement, but it does not provide a meaningful discussion of limitations. In particular, it does not sufficiently discuss:
- the gap between the stylized auction / learning motivation and the proved mathematical model,
- the practical scalability of the proposed PTAS procedures as a function of epsilon,
- the fact that the empirical evidence is limited to small Monte Carlo illustrations,
- the strong Gaussian assumptions underlying all formulations.

A more candid limitations section would improve the paper.

**Strengths And Weaknesses:**

Soundness:
The paper asks a mathematically interesting question and the overall decomposition of the problem is sensible. The distinction between the single-set and multi-set settings is clear, and the combination of a structural lemma for low-variance coordinates with grid search and submodular approximation is natural. I also think the concentration intuition in the dense multi-set regime is interesting, and the connection to submodularity in the equal-variance restricted case is a useful algorithmic observation.

However, I have several concerns about correctness and rigor in the current version.

First, there are concrete inconsistencies in the formal statements. For example, the problem formulation for CorrVarAlloc allows nonzero means mu_i, but Theorem 1.2 is stated for X distributed as N(0, Sigma_hat), while Algorithm 2 is described as handling arbitrary means. The theorem statement, algorithm, and proof should agree on whether the correlated PTAS is for zero-mean or arbitrary-mean Gaussians.

Second, the pseudocode for the PTAS algorithms omits essential feasibility constraints. In Algorithm 1, the grid search over sigma values does not explicitly enforce the budget constraint sum_i sigma_i^2 = 1. In Algorithm 2, the search over covariance matrices checks positive semidefiniteness but does not explicitly enforce trace(Sigma) = 1. Since these constraints define the optimization problem, omitting them from the algorithm descriptions is a serious issue.

Third, there are visible mathematical inaccuracies in the exposition. On page 2, the lower bound used to argue that an additive PTAS is at least as strong as a multiplicative PTAS appears incorrect: if one sets sigma_1 = 1 and all other variances to zero, then the expectation of max_i X_i with additional zero variables is E[max(X,0)], which is 1 / sqrt(2 pi), not sqrt(2 / pi). This is not the main theorem, but it raises concern about the level of checking.

Fourth, the learning and auction motivation is highly stylized and does not yet cleanly justify the exact formal optimization problems. In particular, the mapping from controllable training weights or OU parameters to an arbitrary covariance-allocation optimization problem is suggestive, but not derived in a fully convincing way.

Presentation:
The paper has a clear high-level structure and the main variants of the problem are introduced early. I appreciated that the authors separate the single-set PTAS results from the multi-set logarithmic approximation results, and the overview section helps the reader understand the intended logic of the proofs.

That said, the presentation still needs substantial work. There are many notation issues, theorem-algorithm mismatches, and proof details that are left too implicit. Several statements that should be precise are currently informal. For example, the algorithm descriptions omit crucial constraints, the correlated and non-correlated cases are not always cleanly separated, and the modeling section mixes motivation and formalism in a way that makes it hard to tell what is actually proved versus what is only heuristic interpretation.

I also found the empirical section weak as evidence for the algorithmic claims. Figures 1 and 2 are only small Monte Carlo illustrations on Erdos-Renyi-style instances and simple block-diagonal correlation patterns; they are useful as intuition, but they do not meaningfully validate the approximation algorithms or the practical relevance of the proposed objectives.

Significance:
The optimization problem itself is interesting, and I can imagine it being useful to researchers working on Gaussian processes, stochastic optimization, or submodular-style approximation methods. The single-set PTAS claims and the structural lemma are potentially meaningful theoretical contributions if they are fully correct and carefully written.

For an ML venue, however, the significance currently feels limited. The connection to machine learning is mostly motivational, there are no realistic ML experiments, and the algorithmic evaluation is confined to small illustrative simulations rather than empirical tests of the proposed methods on learning or auction tasks. As a result, the paper currently reads more like a theory note with a speculative ML motivation than a strong ICML submission.

Originality:
I do think there is originality in the exact way the paper formulates variance allocation under Gaussian-max objectives and in the way it combines a chaining-based structural lemma, grid discretization, and submodular approximation ideas across the different variants. The single-set versus multi-set split is well chosen, and the concentration story for dense multi-set instances is also interesting.

At the same time, some components are closer to standard techniques: discretization for PTAS design, reduction to submodular maximization, and Wasserstein / Earth-Mover continuity arguments are familiar tools. So I view the work as moderately original overall, with novelty coming more from the problem formulation and combination of ideas than from one clearly dominant technical breakthrough.

---

> ### Author Rebuttal · Authors · 2026-03-29
>
> Thank you for your thorough review. We totally agree with your findings and will fix the typos and mathematical inaccuracy in the final version. We will also add an empirical section as suggested. Here we address your questions with the empirical result first, and the other questions follow.
>
> **1. Experiment of our algorithm:**
>
> While our paper focuses on a theoretical model of variance allocation, it is deeply motivated by an important practical online advertising setting in major search platforms, where advertisers bid on keywords.  In real-world online auctions, most keywords only have very few bidders involved (can be as few as 2~4). We add an experiment below simulating first-price auctions on sparse uniform hypergraphs (\texttt{GraphVarAlloc}) to test our algorithm (i.e. Algorithm 3) and see whether its approximation ratio is really $\Omega(\log n)$ for independent bidders.
>
> - **Setup:** 2-4 competing bidders per auction with means $\mu_i \sim U[0, 1]$ or $\mu_i = 0$. Variance budget is 1; $n \in \{10,20,50\}$. Bidders appear randomly in 4 or 8 sets. Averaged over 100 runs.
>
> - **Benchmarks:** (1) *Uniform allocation* ($\frac{1}{\sqrt{n}}$ variance per variable). (2) *SLSQP*, a local optimal solver for sum-of-squares constrained optimization. Note: for 2-bidder, 0-mean auctions, SLSQP is globally optimal (since objective is concave per Proof of Theorem 2.5).
>
> - Expectations of max of Gaussian are computed via Gauss-Hermite Quadrature, improving upon Monte-Carlo simulation.
>
> **Results**
>
> $\mu_i=0$
>
> |$k$|$d$|$n$|Uniform|SLSQP|Alg 3|Alg 3/SLSQP|
> |-|-|-|-|-|-|-|
> |2|4|10|3.57|3.90|3.60|92.24%|
> |2|4|20|5.05|5.75|5.33|92.84%|
> |2|4|50|7.98|9.12|8.37|91.75%|
> |2|8|10|7.14|7.23|6.12|84.54%|
> |2|8|20|10.09|10.68|10.31|96.48%|
> |2|8|50|15.96|17.23|15.03|87.24%|
> |3|4|10|3.48|3.90|3.63|93.11%|
> |3|4|20|4.92|5.63|5.20|92.29%|
> |3|4|50|7.90|9.09|8.29|91.18%|
> |3|8|10|6.96|7.41|6.68|90.05%|
> |3|8|20|10.03|10.84|10.34|95.40%|
> |3|8|50|15.92|17.38|15.23|87.62%|
> |4|4|10|3.26|3.64|3.43|94.27%|
> |4|4|20|4.60|5.37|4.91|91.59%|
> |4|4|50|7.28|8.58|7.96|92.73%|
> |4|8|10|6.51|6.96|6.40|91.96%|
> |4|8|20|9.21|10.11|9.55|94.47%|
> |4|8|50|14.56|16.05|14.37|89.55%|
>
> $\mu_i\sim U[0,1]$
>
> |$k$|$d$|$n$|Uniform|SLSQP|Alg 3|Alg 3/SLSQP|
> |-|-|-|-|-|-|-|
> |2|4|10|15.07|15.62|15.55|99.54%|
> |2|4|20|28.56|29.47|29.48|100.02%|
> |2|4|50|68.68|70.04|70.34|100.44%|
> |2|8|10|29.78|30.19|30.17|99.90%|
> |2|8|20|57.84|58.91|58.88|99.94%|
> |2|8|50|136.38|138.20|138.48|100.21%|
> |3|4|10|11.30|12.05|11.98|99.41%|
> |3|4|20|21.30|22.41|22.46|100.19%|
> |3|4|50|51.38|52.98|53.36|100.74%|
> |3|8|10|22.52|23.60|23.51|99.63%|
> |3|8|20|43.12|44.67|44.71|100.09%|
> |3|8|50|102.80|105.22|105.75|100.50%|
> |4|4|10|9.43|10.28|10.21|99.26%|
> |4|4|20|17.60|18.87|18.90|100.16%|
> |4|4|50|42.01|43.88|44.16|100.62%|
> |4|8|10|18.78|20.14|20.07|99.62%|
> |4|8|20|35.06|37.06|37.05|99.98%|
> |4|8|50|83.43|86.31|86.79|100.55%|
>
> **Observations**
>
> Empirically,
> - Alg 3 does not scale as poorly as $\Omega(\log n)$. It consistently beats uniform allocation and performs near-optimally compared to the local optimal SLSQP for all cases (even outperforming it for random means).
> - Runtime of Alg 3 is similar to that of SLSQP (both ~1-4 sec on our instances).
>
> **2. Scope of Theorem 1.2:**
> Thank you for identifying the inconsistency. The problem formulation and Algorithm 2 handle arbitrary means, but theorems in the intro mistakenly stated 0 mean. Throughout the paper, we put a lot of effort into making sure that the theorems work for arbitrarily non-negative means. We will update the theorem statements and emphasize the versatility of the settings in the intro accordingly.
>
> **3. Feasibility constraints in Algorithms:**
> The constraints are currently captured by a soft statement like "For all possible $(\sigma_{i_1},\cdots,\sigma_{i_k})$...", and we will clarify this in the final version.
>
> **4. Typo on Page 2:**
> You are definitely correct about this and we will correct this in the final version. As $1/\sqrt{2\pi}=\Omega(1)$ still holds, the optimal objective remains bounded away from $0$, preserving the logic needed in the paper.
>
> **5. Computing max of Gaussian:**
> This definitely needs to be explained in the paper. Although the exact computation of the expectation of the max of Gaussian is hard, approx with $\epsilon$-accuracy can be solved efficiently. Raghu Meka's paper "A Polynomial Time Approximation Scheme For Computing the Supremum of Gaussian Processes" shows how to efficiently compute the sup of Gaussian processes with $\epsilon$ error, which implies OBJ (expectation of max of correlated Gaussian) can be evaluated efficiently with $\epsilon$ loss.
>
> **6. Asymptotic assumptions in Theorem 1.6:**
> We clarify that $p$ can depend on $n$, provided $p = \Omega(1/n)$ (so the sets of variables do not become trivially empty).
>
> **7. On discussion of limitation:**
> Thank you for listing the limitations, we will add a discussion section as well.
>
> Please consider raising the score if you see fit.

---

### Official Review · Reviewer_b2Sv · 2026-03-13

**Soundness:** 2
**Presentation:** 2
**Significance:** 2
**Originality:** 3
**Overall Recommendation:** 4
**Confidence:** 3

**Summary:**

This paper addresses the approximation problem of maximizing the expectation of the supremum of families of Gaussian random variables. Concretely, they focused on four problems (VarAlloc, CorrVarAlloc, GraphVarAlloc, and CorrGraphVarAlloc) and proposed different algorithms to (approximately) solve them with theoretical guarantees.

**Compliance With Llm Reviewing Policy:**

Affirmed.

**Final Justification:**

The paper studied an interesting problem in my view. Given the authors' reply to my questions and additional experiments, I will increase my score from 3 to 4. However, the authors should carefully revise the paper to address the issues identified in the review.

**Key Questions For Authors:**

This question will not affect the paper too much. But I also wonder whether the constant $2e/(e-1)$ in Lemma 2.8 can be further improved? Or it is already tight.

**Limitations:**

yes

**Strengths And Weaknesses:**

**Strengths.**

Overall, the paper considered different interesting problems and proposed different solutions to address them.

**Weaknesses.**

1. My main concern is about the proof of Lemma 2.4, in which the authors stated $d_{EMD}(D_1,D_2)\leq d_{W_2}(D_1,D_1)$ (Line 748). However, I cannot see why this step holds, since $d_{EMD}$ is currently defined by the $\ell_1$ norm, whereas $d_{W_2}$ is defined by the $\ell_2$ norm.

1. Lemma 2.2 (A.2) seems to hold only under some (in)dependence requirement, since it relies on Lemma A.3.

1. I think the proof of Lemma A.3 can be largely simplified by assuming $X=\mu+\sigma_1 Z$ and $Y=\mu +\sigma_2 Z$ (where $Z$ and $T$ are independent) and using $|\max(X,T)-\max(Y,T)|\leq |X-Y|=|\sigma_1-\sigma_2||Z|$.

1. There are also some typos. For example:

   1. "an set" should be "a set".

   1. In many places, "GraphVarAllocwith" should be "GraphVarAlloc with". A space is missing.

   1. “$E[$...” should be "$\mathbb{E}[$...".

   1. In the statement of Lemma A.3, $(\mu,\sigma_1)$ (resp. $(\mu,\sigma_2)$) should be $(\mu,\sigma_1^2)$ (resp. $(\mu,\sigma_2^2)$).

   1. “PDF X” should be "PDF of X".

---

> ### Author Rebuttal · Authors · 2026-03-29
>
> Thank you for your rigorous review, and we are especially grateful for your elegant proof simplification. We will fix the typos in the final version. Regarding your other specific points:
>
> **1. On $L_1$ vs $L_2$ norm mismatch in Proof of Lemma 2.4:**
> You are absolutely correct that we define EMD using the $\ell_1$ metric, whereas the $W_2$ distance is defined by the $\ell_2$ metric. To fix this, we can use the correct inequality -- $\|x\|_1 \le \sqrt{k}\|x\|_2$, resulting in a $\sqrt{k}$ factor loss in the lemma statement. This dimension factor $\sqrt{k}$ does not break our PTAS result: by our structural lemma, we only assign non-zero variance to at most $poly(1/\epsilon)$ variables. Thus we can simply tighten the grid search precision while preserving the PTAS guarantee and the polynomial running time. We will update the proof to make this correction explicit.
>
> **2. On independence requirement in Lemma 2.2:**
> We agree that the lemma statement itself should be self-contained and does not require an implicit assumption in the current section. We will add the assumption in the lemma statement.
>
> **3. On simplification of Lemma A.3:**
> We appreciate your elegant proof simplification. Following your suggestion we no longer need to prove this intuitively correct statement with a lengthy and brute-force integral derivation. We will replace the old proof with your elegant approach and credit your review explicitly.
>
> **4. On improving the constant in Lemma 2.8:**
> While sufficient for our asymptotic logarithmic approximation, we don't believe that the $\frac{2e}{e-1}$ gap is tight. We will leave this as an open question.
>
> We have also added additional experiments (empirical study on the actual approximation factor achieved by some of our algorithms) in response to comments by Reviewer vkB2. Please see that response. We hope that our responses above and the additional experiments help  address your concerns about our paper.

---

> > ### Author Rebuttal · Reviewer_b2Sv · 2026-04-02
> >
> > Thanks for the response. I would like to believe the authors can fix the theoretical issue I mentioned and also revise the paper carefully. Given that, I will increase the rate to 4.

---

### Decision · Program_Chairs · 2026-04-30

**Decision:**

Accept (regular)

**Comment:**

This paper studies the problem of allocating variance (or covariance) to maximize the expected supremum of Gaussian random variables, providing a PTAS for the singleton setting and an O(log n) approximation for multi‑set variants, with additional structural results on variance concentration.

The scores are: 4 (Weak accept), 3 (Weak reject), 5 (Accept) — overall positive. In the rebuttal, the authors addressed technical concerns, added new experiments validating the algorithms, and clarified the scope of their theorems; Reviewer b2Sv increased their score to 4, Reviewer 6vyU maintained accept, and Reviewer vkB2 (Weak reject) didn't acknowledge the rebuttal and additional exp results (hence this rating is down-weighted).

Given the theoretical contributions and the fact that the majority of reviewers support acceptance, I recommend acceptance.